



# Quantifying evaporation of intercepted rainfall: a hybrid correction approach for eddy-covariance measurements

Stefanie Fischer[1], Ronald Queck[1], Christian Bernhofer[1], and Matthias Mauder[1]

[1]Technische Universität Dresden, Institute of Hydrology and Meteorology, Department of Meteorology, Pienner Str. 23, 01737 Tharandt, Germany

**Correspondence:** Stefanie Fischer (stefanie.fischer@tu-dresden.de)

**Abstract.** Precipitation and interception have a significant influence on the reliability of eddy-covariance (EC) measurements, primarily of vapor fluxes. As evaporation data need to fit both to the energy and the water budget, a balanced approach is necessary to arrive at reasonable values of evaporation associated to interception. EC data of the investigated *ICOS* site DE-Tha (dense conifers) suggest a large and systematic underestimation of evaporation during and shortly after a rainfall event. Total

evaporation of selected interception events accounted for only 24% of precipitation, which is an untypically low proportion for a dense coniferous forest under a temperate climate. We show that our Rutter based 2D model approach, including spatially variable vegetation information, reproduces reliable estimates of interception evaporation to compare and integrate the results for different source areas. For the EC footprint area, modelled interception evaporation accounts for 45% of precipitation for the evaluated events. The standard Bowen ratio based energy balance adjustment and the energy balance residual approach are

not justified to account for underestimated fluxes during interception events. As a consequence, we propose a hybrid correction approach complementing EC measurements with our 2D model estimates of evaporation under interception conditions to adjust for underestimated fluxes of LE. Our approach uses LE as a link between the energy and water balance and provides appropriate evaporation from intercepted precipitation for the analyzed forest ecosystem. The correct redistribution of the heat fluxes will lead to a better parametrization of surface fluxes in weather and climate models and supports to properly include

land use in water management needs under climate change.

## 1 Introduction

Land surface-atmosphere interactions, such as the exchange of energy and water are largely controlled by vegetation. A key component of this exchange is evaporation, which entangles several distinct processes such as interception $E_I$, transpiration $E_T$, soil evaporation $E_S$ and evaporation from open water $E_O$ together summing up to the total flux of water vapor into the

atmosphere $E_{tot}$.

$$E_{tot} = E_T + E_I + E_S + E_O \tag{1}$$

Many advocate to analyze these individual processes distinctively (Savenije, 2004; Miralles et al., 2020) as they differ in terms of time scale, time of occurrence, physical characteristics and atmospheric feedback. About 62% of precipitation





occurring at the Earth's terrestrial surfaces is evaporated. The contribution of transpiration to $E_{tot}$ is between 25 to 64%, only
about 3% is due to open water evaporation and most of the remainder is due to interception, while soil evaporation contributes
only a very small amount to the total (Dingman, 2015). Hence, transpiration and evaporation of intercepted water are the major
processes with respect to the total evaporative flux over land. Both scale roughly with the plant surface area, emphasizing
the role of vegetation. The proportion of either process depends not only on vegetation properties, but also on atmospheric
conditions and rainfall characteristics (Stewart, 1988; Lian et al., 2022). Comparing, for instance, both components for tall
and rough forests, which are strongly coupled to the atmosphere, evaporation rates of intercepted precipitation are several
times higher than water fluxes from transpiration under similar atmospheric conditions (Rutter et al., 1971; Stewart, 1977;
Teklehaimanot and Jarvis, 1991).

Until around 1990, there was no standardized measurement method to quantify the total evaporation of ecosystems for all
important types of land use. The only exception was the catchment approach, which derives long term evaporation from the
hydrological budget via the difference of precipitation and runoff. However, this indirect method only allows the determination
of evaporation over periods of several years, depending on the underground storage in the catchment area. Since then, the direct
measurement of the turbulent transport by eddy covariance began to fill this gap. Over the past 25 years, hundreds of sites have
been established across the globe to measure carbon fluxes and evaporation from all kinds of terrestrial surfaces, with the data
being coordinated and standardized by several regional or global observation networks such as *ICOS* (Integrated Carbon Obser-
vation System, https://www.icos-cp.eu) and *NEON* (National Ecological Observatory Network, https://www.neonscience.org/)
or common data portals such as *FLUXNET* (https://fluxnet.org) (Baldocchi et al., 2001). While this is a success story, showing
that a world wide data set on ecosystem-scale fluxes can be made available, the eddy covariance data itself require complex se-
tups, data handling (esp. post processing) and careful interpretation. The most discussed source of uncertainty is the insufficient
energy balance closure EBC, probably due to underestimation of the total flux by EC measurements which measure turbulent
fluxes only (Mauder et al., 2024). Despite the identification of various factors contributing to the observed energy imbalance,
it remains unclear to which extend the two main components of evaporation from vegetation - transpiration and interception -
are affected by the problem.

Generally, turbulent transport is often not covering the total flux, instead advection and dispersive fluxes are prominent.
Approaches correcting for the unaccounted energy (Mauder et al., 2018; Bambach et al., 2022; Zhang et al., 2023, 2024;
Wanner et al., 2024) typically do not distinguish between transpiration and evaporation from intercepted rainfall. A large
proportion of the imbalance and one of its main reasons can be attributed to energy transport through secondary circulations.
This is associated to dispersive fluxes, a phenomenon restricted to well coupled, unstable stratification and prevalent in daytime
convective boundary layers (Mauder et al., 2021; Bambach et al., 2022; Wanner et al., 2024). However, these conditions are
rather relevant for dry weather conditions associated with transpiration. In contrast, interception episodes are characterized by
stable stratification, with a downward directed heat flux and suppressed or mechanical turbulence. EC measurements during
these conditions poorly cover vertical transport and are questionable, since the role of a "wet-bulb effect" (horizontal advection
from dry areas), low-frequency turbulence and infrequent large-scale coherent structures remains unknown (van Dijk et al.,
2015; Stoy et al., 2019; Fischer et al., 2023).





Reliable data including uncertainty across sites and climates will contribute adequately to a better understanding of the complex processes of evaporation. The problem of an incomplete energy balance needs to be addressed, as EC data serves as input to parameter estimates in models at all scales from plot to global, especially for *Soil Vegetation Atmosphere Transfer* schemes (Falge et al., 2005). This fosters research of ecosystem-atmosphere interactions and allows a consistent modelling of water and heat fluxes. Here, precipitation, energy and evaporation partitioning play a major role to understand the effect of changing vegetation systems or climatic conditions on the budgets of water and energy (Lian et al., 2022; Pluntke et al., 2023; Zhang et al., 2023).

Hence, there is a need to validate EC based estimates of evaporation under variable weather conditions by independent measurements. Particularly, the role of interception is understudied. However, the source area that contributes to the measured fluxes (footprint area) using the eddy covariance technique is dynamic in extend and direction due to its dependency on wind direction and atmospheric stratification (Foken et al., 2005; Chu et al., 2021). Thus, accurate validation is complicated due to the varying spatial and temporal scales inherent in the underlying measurement concepts, including lysimeters, sap flow and canopy water balance. An additional challenge poses the partitioning of total evaporation (Yi et al., 2024).

The objectives of this study are to arrive at a consistent data set of water and heat fluxes. This data set should fulfill the energy and water budget with as little compromises as possible. To achieve this goal, we need to address the following points. Site homogeneity is a fundamental assumption for the simple use of EC data, i. e. the homogeneity of the underlying exchange "surface". In reality, it is a volume with complex three dimensional structured vegetation, including also soil, biomass, air and water (Schmid, 1997). This study considers the spatial extent of the flux footprint area, particularly in wet weather conditions, and investigates how well this is reflected by independent estimates of interception. The approach requires a well known canopy structure and a suitable high resolving canopy model. Regarding this, we introduce a grid based application of the Rutter approach (Rutter et al., 1971). The model enables the incorporation of forest stand structure, derived from terrestrial laser scanning *TLS*, to consider the two dimensional spatial heterogeneity of the vegetation. This allows a spatially variable water and energy partitioning depending on plant area to demonstrate the associated spatio-temporal variability of the interception process. At the example of the *ICOS* site Tharandt Forest (DE-Tha, Norway spruce) and the application of the 2D Rutter approach, we

(i) compare two separate data streams for events of interception, the measurement of gross precipitation minus throughfall (classical canopy water balance; WB method) and the simultaneous EC data (EC method),

(ii) assign the EC data to a source area allowing to consider stand heterogeneity, including the spot measurements by the WB method,

(iii) identify the best available method to adjust and gapfill the EC measurements of latent heat flux under interception conditions

(iv) assess the effect of the adjustment on the overall water budget and the partitioning of precipitation in the study area.



## 2 Material and methods

### 2.1 Study Site and setup

The study site Tharandt *DE-Tha* (50.96235°N, 13.56529°E, 385 m a.s.l.) is part of the Integrated Carbon Observation System *ICOS* (https://meta.icos-cp.eu/resources/stations/ES_DE-Tha) and equipped with a well maintained environmental research
structure to provide meteorological, hydrological and ecological measurements. It is located in the Tharandt Forest (60 km$^2$), which is situated in the lower Eastern Ore Mountains about 25 km southwest of the city of Dresden (Germany). The first national monitoring network dedicated to forest-meteorological observations is associated with the Tharandt Forest. The network also happens to be the location with the first reported direct observations of throughfall in the mid-nineteenth century (Van Stan et al., 2020).

The forest at the research site is subject to standard management and is mainly characterized by evergreen conifers. A spruce stand was established in 1887. Today, it consists of 72% Norway spruce (*Picea abies*) and 15% Scots pine (*Pinus sylvestris*). The remaining canopy is composed of deciduous trees with 10% European larch (*Larix decidua*), 1% birch (*Betula spec.*) and 2% other species using data from 1999 as a reference (Grünwald and Bernhofer, 2007). The more recent stand is described in detail by Queck et al. (2016) with a dense canopy (335 trees ha$^{-1}$), a mean tree height of 31 m and an open trunk space with
sparse understory for the year 2008. Continuous in-canopy radiation measurements yielded a leaf area index LAI value of 7.1 around the flux tower for the same year. The general characteristics of the canopy have not changed significantly for 25 years. Grunicke et al. (2020) concluded in a long-term interception study that changes in stand structure, density and LAI remained relatively small and less variable for the period 2008 to 2018. The canopy can be characterized as sufficiently homogeneous for the source area of the throughfall measurements. However, the statistic representation of rainfall partitioning for the footprint
of the EC flux measurements by the throughfall collection plots in the vicinity of the tower has not been analyzed in detail so far.

Direct flux measurements (including sensible $H$ and latent heat $LE$) are carried out above the forest canopy in 42 m above ground together with a multitude of meteorological measurements (global and net radiation $R_g$ and $R_n$, air temperature $T_{air}$, relative humidity $rH$ and wind speed $u$) at various heights of the tower. A detailed summary of the instrumentation and data
quality can be found in Fischer et al. (2023). The collection of throughfall and gross precipitation is described in detail by Grunicke et al. (2020). In summary, two large trough systems of 10 m length each and a total area of 3.18 m$^2$ are collecting throughfall with a time resolution of 10 minutes. The measurements are restricted to periods without frost, typically from April to September. Gross precipitation is measured with a resolution of 10 minutes about 130 m west of the flux tower in a forest clearing of 50 x 90 m called *Wildacker*, which is also the location of the meteorological station. Warm summers with mostly
convective precipitation and cold winters with weaker frontal rain or snow are characteristic for the sub-continental climate of the study site. Mean annual temperature is 8.7°C and mean annual sum of precipitation is 842 mm a$^{-1}$ based on long-term records for the period 1991 to 2020.



All data used in this study refer to the period 2008 to 2010, since a detailed 3D representation of the forest is available as a result of terrestrial laser scans during these years (Bienert et al., 2010). A map of the study site with vertically integrated $PAD$
($PAI_{Local}$) of the area is shown in Figure 1.

## 2.2 Canopy water balance

The classical canopy water balance WB serves as a reference to quantify interception for the study site *DE-Tha* (Eq. 4). Here, data of gross precipitation $P_g$ measured in a forest clearing was gapfilled and corrected by independent rainfall measurements of daily resolution (Grunicke et al., 2020). Stemflow $T_s$ is considered to be negligible for the dominating spruce trees in the
research area due to their bark structure and architecture (Rutter et al., 1975; Cisneros Vaca et al., 2018; Grunicke et al., 2020). Forest floor interception $E_{I,FF}$ is difficult to measure as it is very heterogeneous on the small scale. Gerrits et al. (2010) summarize the importance of $E_{I,FF}$ and that it might reach to the same amount as canopy interception. However, evaporation of intercepted water from the litter takes longer than that from the canopy, thus we assume that the forest floor evaporation during an event is negligible. Thus, measurements of free throughfall $T_f$ and drainage $T_d$ (above the litter layer of the forest
floor) are representing net precipitation $P_n$, allowing to calculate interception $E_{I,WB}$ from the canopy water balance (WB) as the residual of $P_g$ and $T_f$.

$$P_g = T_f + T_D + T_s + E_{I,WB} + E_{I,FF} \tag{2}$$
$$\approx T_f + T_D + E_{I(WB)} \tag{3}$$
$$\approx P_n + E_{I(WB)} \tag{4}$$

$E_{I,WB}$ is analyzed as the sum based on events and no attention is given to the dynamics on a sub-event scale. Only data with liquid rainfall during frost free periods was analyzed and all events with mean temperatures less than 2°C and minimum temperatures lower than 0°C were excluded from the analysis.

## 2.3 Eddy covariance data

**LE from Measurement and Postprocessing $LE_{EC}$:** Turbulent fluxes of sensible $H_{EC}$ and latent heat $LE_{EC}$ are measured
via eddy covariance technique by the use of an ultrasonic anemometer (SA-Gill-R3-50) and a closed-path infrared gas analyzer (LI-COR-LI7000). The *ICOS* processing chain according to Sabbatini et al. (2018) was used. Post-processing for post-field raw data and quality control was done with EddyPro®. Typical situations occurring under rain or wet canopy conditions have been addressed in the data processing, such as potential signal loss of water vapor fluxes due to tube attenuation or sensor separation (Fratini et al., 2012) and the detection of records with weak variance during stable conditions or low wind speeds (Vickers and
Mahrt, 1997). Additionally, the model after Kljun et al. (2015) was applied to calculate the extent and relative contribution of a source area to the total flux measurement. The spatial extents of the 30-min flux footprints with a relative flux contribution up to 90% were calculated for the duration of each modelled frost free interception event. The flux data $F$ was gapfilled together





with meteorological measurements and the quality of the approach was tested. The procedure is explained in detail in Fischer et al. (2023).

**LE from Energy Balance** $LE_{EB}$**:** The sum of all fluxes $\Sigma J$ associated with heat stored or released in the moist volume of air below the forest's canopy, within the ground or vegetation has been calculated. Finally, latent heat flux as the residual of the energy balance ($LE_{EB}$) was calculated using Eq. 6. This approach attributes the systematic error in the flux measurements by the EC method entirely to latent heat $LE$ closing the energy balance. The first term on the left hand side of Eq. 6 in brackets, encompassing net radiation $R_n$, ground heat flux $G$ and the sum of all storage fluxes $\Sigma J$, corresponds to all available energy sources that drive the turbulent fluxes and is therefore called available energy $AE$. Further details can be found in Fischer et al. (2023).

$$LE_{EB} = (Rn - G - \Sigma J) - H \tag{5}$$
$$= AE - H \tag{6}$$

**LE corrected by Bowen ratio** $LE_{\beta}$**:** Another approach to account for the underestimation of the total flux by EC measurements, is the redistribution of the energy balance residual to both fluxes $H$ and $LE$ according to their Bowen ratio ($\beta = H/LE$). In this study, the approach after Mauder (2013) was applied, in which flux data is corrected for daytime conditions with global radiation $R_g > 20\mathrm{W\,m^{-2}}$. Hence, this method accounts for the systematic error related to a convective boundary layer, which allows the development of thermally driven large scale and non-propagating eddies.

$$EBR = \frac{\sum_{i=1}^{K}(H_{EC} + LE_{EC})}{\sum_{i=1}^{K} AE} \tag{7}$$

$$F_{\beta} = \frac{F_{EC}}{EBR} \tag{8}$$

The energy balance ratio (EBR) is calculated on a daily basis for half-hourly fluxes, restricted to situations with $R_g > 20\mathrm{W\,m^{-2}}$ (Eq. 7). For nighttime conditions EBR is set n/a. The respective flux $F$ measured by the EC method ($H_{EC}$ and $LE_{EC}$) is finally adjusted according to Eq. 8. Resulting outliers were removed using the $4\sigma$-filtering method. All half-hourly gaps of $LE_{\beta}$ and $H_{\beta}$ were filled by the use of the software package REddyProc (Wutzler et al., 2018).

## 2.4 2D Rutter Model

The conceptual framework after Rutter et al. (1971) was applied calculating the canopy water balance for the spruce forest dynamically. Two model setups were employed to account for spatial variable vegetation characteristics in the study area. In a basic approach, the whole vegetation stand of the study area was modelled as one big leaf with a mean PAI of $4.65\mathrm{m^2\,m^{-2}}$ (please note that this is somewhat smaller than the PAI given in the description of the study site, as it is not the PAI around the measurement tower but for the long term footprint area of the EC measurements). The second model did describe the





horizontal variability of the vegetation with a resolution of 10m in a gridded domain of $x = 1140\text{m}$ and $y = 800\text{m}$. Both of the models are big leaf models that use no vertical differentiation over the entire height of the vegetation (33m). The vegetation structure was captured by airborne laser scanning and terrestrial laser scanning (see Bienert et al. (2010); Queck et al. (2012)). From the work of Queck et al. (2016) a vegetation model with a spatial resolution of one cubic meter was available which was

integrated to gain the 10m resolution. The resulting spatially variable $PAI_{Local}$ of the study domain is presented in figure 1 with an overall $PAI$ of $4.65\text{m}^2\,\text{m}^{-2}$. Processes and model parameters such as partitioning of precipitation and evaporation components, drainage or canopy storage capacity, are calculated as a function of PAI. Storage depletion is simulated by an exponential drainage approach and by evaporation, which is calculated based on the Penman-Monteith (*PM*) equation. This results in a water and energy budget for each grid cell depending on the distribution of vegetation (PAI) in the study area.

Model structure, parameters and main process equations are explained in detail in the appendix A.

The meteorological input included a time series of wind speed $u$, water vapor pressure $vpd$ and air temperature $T_{air}$ in 33m height, as well as net radiation $R_n$ and global radiation $R_g$ in 37m height. The model simulations were conducted for the period from 2008 to 2010. The resulting events of interception were filtered for liquid rainfall conditions with frost free periods (only events with mean and minimum temperatures higher than $2°\text{C}$ and $0°\text{C}$, respectively). Time steps between modelled and

measured data (EC and 2D approach) were synchronized for comparison. The grid of the 2D simulation results was assigned to the respective footprint areas of the EC measurement and to the area of the throughfall collectors (WB approach). The model performance was evaluated by the coefficient of determination $R^2$, relative absolute error $RAE$ (Eq. 9) and Nash-Sutcliffe-Efficiency $NSE$ (Eq. 9) for measured (obs) and modelled (mod) throughfall (with $\mu$ as the average value).

$$RAE = \frac{\sum (|obs - mod|)}{\sum (|obs - \mu_{obs}|)} \tag{9}$$

$$NSE = 1 - \frac{\sum (|obs - mod|)^2}{\sum (|obs - \mu_{obs}|)^2} \tag{10}$$

## 3   Results

### 3.1   Model validation

The 2D model was evaluated on event basis by canopy interception $E_{I,WB}$ (Eq. 4), which is retrieved according to the WB approach described in section 2.2. Modelled events were filtered for liquid rainfall conditions (frost free periods), for which

the reference measurements of the canopy water balance approach (WB) are reliable. Additionally, only events with footprints fitting inside the model domain were selected. Figure 2 shows the 2D model results for i) the location of the throughfall collectors (WB location), ii) the dynamic flux footprint area (footprint) and the iii) whole model domain (2D domain). Additionally, the simple big leaf approach for the overall PAI of the grid cells covering the area of the throughfall collectors is depicted in the last panel on the right (big leaf) of Figure 2.





For the majority of events, modelled sums of interception evaporation are agreeing very well with the observations $E_{I,WB}$. The 2D model of the WB location is showing the best agreement with the observed data (NSE=0.85 and RAE=0.21), due to matching source areas. However, a view events exceeding evaporation sums of around 8mm are underestimated by the model, this and the scatter of larger events lead to an overall slope of 0.87. With increasing source areas covering the respective footprints of the EC measurement or the whole model domain, modelled sums of interception evaporation are decreasing,

yielding slopes of 0.78 and 0.73, respectively. Due to a higher overall PAI of $6.54\mathrm{m}^2\,\mathrm{m}^{-2}$ in the WB location, a higher amount of precipitation can be stored on and evaporated from the canopy surface, than for the whole model domain with an overall PAI of $4.65\mathrm{m}^2\,\mathrm{m}^{-2}$. These results indicate, that the spruce forest of the study area cannot be considered homogeneous and that the WB approach does not reflect the rainfall interception for the wider area around the EC tower.

Modelled evaporation from interception of the simple big leaf approach for the WB location (Figure 2, right) agrees only

well with observations for events up to 3mm. This amount corresponds exactly to the storage capacity of the model and the transition from an unsaturated to a saturated canopy leads to an underestimation by the big leaf approach. For events with a precipitation exceeding canopy saturation, modelled throughfall is overestimated leading to underestimated amounts of $E_I$. Results might be improved by adjusting the storage capacity of the model. However, the big leaf model also leads to a higher amount of total interception events (n=302) due to faster drying. In the 2D model approach a variety of grid cells with different

PAI is considered, leading to spatially variable storage, throughfall and drainage and finally the interception event ends with the last cell being dry. Unlike the big leaf approach, this reflects more on the process of precipitation routing within the canopy, which leads to a better fit to observed $E_I$.

**Table 1.** Average evaporation sums ($E_{tot}$) and its components throughfall ($T_f$), interception evaporation ($E_I$) and transpiration ($E_T$) for selected events over the years 2008 to 2010. On average, 50±9 interception events where evaluated per year within this period (only events without frost or snow, with the extend of the EC footprint area inside the model domain) with an average precipitation sum of 146.80±13.34mm. Measured data refers either to the canopy water balance approach (WB, indicated with [1]) or the eddy covariance system (EC, indicated with [2]). Modelled data refers to the output of the 2D Rutter approach for the grid cells covering the WB location or the respective footprint area of the EC measurements.

|  | Measured | Modelled | |
|---|---|---|---|
|  | 1: WB, 2: EC | WB Location | Footprint |
| $T_f$ in mm | $70.49\pm5.28$[1] | $73.64\pm9.21$ | $81.18\pm9.19$ |
| $E_I$ in mm | $76.30\pm10.73$[1] | $73.16\pm7.97$ | $65.61\pm7.30$ |
| $E_T$ in mm | - | $15.81\pm2.41$ | $13.61\pm2.07$ |
| $E_{tot}$ in mm | $35.33\pm2.19$[2] | $88.97\pm10.09$ | $79.22\pm9.05$ |
| $E_I{:}P_g$ in % | $52\pm4$[1] | $50\pm4$ | $45\pm3$ |
| $E_{tot}{:}P_g$ in % | $24\pm2$[2] | $61\pm5$ | $54\pm5$ |



Table 1 summarizes the selected interception events of the 2D model shown in Figure 2 as average totals over the years 2008 to 2010. On average $50 \pm 9$ events were evaluated for each year, which do not represent an annual budget. Additionally the results of the independent WB approach ($T_f$, $E_I$) and the water equivalent of $LE$ from the EC measurement ($E_{tot}$) are presented. Average annual sums of $T_f$ and $E_I$ for the 2D model of the WB location are comparing very well with the independent WB approach. The 2D model shows a slight overestimation for throughfall of only $3\mathrm{mm\,a}^{-1}$ for the sums of selected events, which in turn leads to a $3\mathrm{mm\,a}^{-1}$ underestimation of $E_I$. As discussed above, the WB approach does not reflect the evaporation conditions of the larger footprint area due to differences in vegetation structure and plant area. Hence, average annual sums of $T_f$ are higher for the respective footprint areas ($81.18 \pm 9.19\mathrm{mm}$), resulting in lower sums of interception evaporation ($65.62 \pm 7.30\mathrm{mm}$).

Due to the high agreement with the WB approach, the physically based 2D model is considered as a good reference for further analyses of evaporation components under interception conditions. Thus, EC-based estimates of $E_{tot}$ are compared to the 2D model estimates of total evaporation for the respective EC footprint areas. The average annual sums of the selected events in table 1 demonstrate that evaporation measured by the EC method under interception conditions ($35.33 \pm 2.19\mathrm{mm}$) shows a systematic underestimation with a slope of 0.41 (regression not shown), compared to the 2D model ($79.22 \pm 9.05\mathrm{mm}$). The lower part of table 1 shows the fraction of precipitation being stored on the canopy as interception ($E_I : P_g$) and the fraction of precipitation contributing to the total evaporative flux ($E_{tot} : P_g$). As only a selection of events is taken into account for the analysis, these fractions are not representative as an annual reference. However, modelled and measured $E_I$ for the WB location account for $50 \pm 4\%$ and $52 \pm 4\%$ of precipitation. Due to lower amounts of interception for the footprint areas, $E_I : P_g$ is lower with a ratio of $45 \pm 3\%$. Total evaporation measured by the EC method under interception conditions accounts only for $24 \pm 2\%$ of precipitation, which is even below the fractions referring to the interception component only. Modelled $E_{tot}$ of the footprint related areas accounts for $54 \pm 5\%$, which again indicates that fluxes measured by the EC method are not being well captured under wet conditions.

If the 2D model results are analyzed to estimate the annual water budget of the spruce forest, $E_I$ for the whole study domain accounts for $28 \pm 3\%$ and $E_T$ for $29 \pm 5\%$ of mean annual precipitation. The interception component $E_I$ contributes $49 \pm 1\%$ and transpiration $E_T$ $51 \pm 1\%$, respectively, to modelled total evaporation $E_{tot}$.

## 3.2 Drivers and components of evaporation

### 3.2.1 Source areas

Figure 3 shows an example event and the spatial distribution of $T_f$, $E_I$, $Rn$ and $H$ for the respective footprint area of the EC measurement. The event starts at 12:00 CET at fourth October 2009 and has a total precipitation of $2.8\mathrm{mm}$. The duration of the interception event is 25h, from the start of precipitation until the canopy is completely dry. Please note, that the color scale in each panel refers to the EC footprint only, which overlays the WB area. Additionally, each panel contains the spatial sums of the respective variables for the EC footprint area (white dot) and the WB location with the throughfall gutters (blue rectangle). Total throughfall in panel a) is higher for the EC footprint area with $1.1\mathrm{mm}$ than for the WB location with $0.9\mathrm{mm}$ due to a





lower plant area on average. The highest sums of $T_f$ occur in the area with the highest flux contribution ($> 60\%$, not shown here) covering the west to north-west direction from the tower up to the forest clearing (*Wildacker*). There, most precipitation reaches the ground within the less dense vegetated areas in the western direction from the tower, the grass covered *Wildacker* and on non-vegetated pathways. The gutters for the throughfall measurements are located at the eastern edge of the footprint

where the relative contribution to the measured fluxes is low ($< 10\%$). As a result of the throughfall distribution, average $E_I$ in panel b) is lower for the flux footprint area (1.7mm) than for the throughfall plot with the collecting gutters (1.9mm), since it covers areas and several paths with low or no vegetation. Spatially variable interception amounts are highest in the north-west direction close to the flux tower, since this area contributes most to the measured EC fluxes and contains dense vegetation. Lowest interception is modelled for the forest clearing, pathways and the more the footprint distributes towards the north-east

and along the edges (lowest flux contribution). Both source areas receive the same amount of energy supplied by net radiation $Rn$, which corresponds to a total water equivalent of 1.5mm. The relative contribution to the total sum of $Rn$, according to the flux contributions measured by the EC system, is illustrated in panel d). The spatial distribution of $Rn$ within the footprint area shows a maximum in the west to north-west direction from the tower, sloping down steadily further in the same direction and dropping steeply towards the east. Another source of energy for the evaporation of intercepted precipitation is supplied by

downward directed sensible heat as shown in panel d). The areas with high amounts of $E_I$ in panel b) are showing a negative or downward directed flux of $H$ in panel d). The total supply of $H$ for the EC footprint accounts for 0.7mm. This "wet bulb effect" is higher for the throughfall collection plot (higher amount of rainfall interception) with a sensible heat supply of 1mm. For areas with low $E_I$ or high $T_f$, such as pathways or the forest clearing, energy from net radiation is not or not entirely used for latent energy transfer. For these grid cells, the remaining energy is being used for convective warming as presented

by a positive or upward transport of sensible heat. The residual energy with a total water equivalent of 0.5mm and 0.6mm ($Rn$-$H$-$E_I$) for the EC footprint and the WB location, respectively, is attributed to soil heat transfer.

### 3.2.2 Water and energy budget related components

The temporally aggregated event discussed above is again presented in Figure 4 during the onset of precipitation, now spatially and temporally aggregated in a half-hourly resolution. The events starts at 12:00 CET with low precipitation of about 0.24mm

in the first hour. At 13:00 CET a higher precipitation pulse of 0.8mm leads to an increase of interception storage $C$ slightly exceeding 1mm. Precipitation pauses from 14:00 to 14:30, followed by three increasing pulses from 15:00 to 16:00 CET, with a total of 1.63mm. While canopy water storage is slightly decreasing due to evaporation with paused precipitation, it increases with recurring $P_g$ to a maximum of 3.1mm. Relative humidity $rH$ is below 70% until the event starts and steeply increases up to 90%. Maximum $rH$ is reached (95%) with the maximum of the canopy storage $C$ and decreases very slowly. Canopy water

storage also decreases slowly, after precipitation has ceased since conditions for evaporation are limited. On the one hand, horizontal wind velocity $u$ decreased with the onset of precipitation, ranging between 1.6 to 2.6m s$^{-1}$ until 23:30 CET. At the other hand, energy is only supplied by net radiation $Rn$ until 15:30 CET. With sunset around 17:00 CET, $Rn$ further decreases from -65.6W m$^{-2}$ to -84.4W m$^{-2}$. Hence, evaporation or heat exchange is only driven by ventilation (vapor pressure deficit and wind) and the energy is supplied by the sensible heat flux or the heat storage.





The lowest panel in Figure 4 shows the turbulent heat fluxes $LE$ and $H$ for the EC measurement (solid line) and the 2D model (dot-dashed line). For the dry conditions up to 12:00 CET, $LE$ shows a similar course for the measurement and the model. Until sunrise (6:30 CET), $LE_{2D}$ is zero and $LE_{EC}$ shows a relatively constant heat flux of about 18W m$^{-2}$. With increasing $Rn$, courses of $LE$ are very similar in magnitude, but the modelled data $LE_{2D}$ follows the course of $Rn$ more distinctively. This highlights the effect of solar radiation on transpiration, which is included in the 2D model as *Jarvis* parametrization of canopy

resistance (Stewart, 1988). The course of sensible heat flux for the dry conditions before 12:00 CET is also similar for the EC measurements and the 2D model. However, fluxes are higher for $H_{2D}$, since the energy balance for the model is closed, which is not the case for the EC data (probably underestimated $H$). Modelled evaporation increases at 11:30 CET with increasing $Rn$ and continues increasing within the first block of $P_g$ (12:00 to 14:00 CET) up to 148.9W m$^{-2}$ in which $Rn$ remains relatively high (97.697.6 to 242.6W m$^{-2}$). The energy supplied by $Rn$ is sufficient to enhance evaporation, which leads to

a decreasing canopy storage $C$ until the next block of precipitation. When $Rn$ falls below 100W m$^{-2}$, additional energy for $LE_{2D}$ is supplied by a downward directed flux of sensible heat $H_{2D}$. After rainfall has ceased, supply of shortwave radiation approaches zero and $Rn$ gets negative. Under these conditions, only advective energy supplied by sensible heat $H_{2D}$ drives evaporation, resulting in a slow and steady decrease of $C$ of 0.04mm h$^{-1}$. This corresponds to a flux density of 26.8W m$^{-2}$ for $LE_{2D}$.

$LE_{EC}$ shows a rather erratic behavior with a sharp decrease (-15.3W m$^{-2}$) when precipitation gets more intense (13:00 CET). After that, $LE_{EC}$ continues to increase up to 69.5W m$^{-2}$, which is close to the modelled result. However, when precipitation continues in the second block, $LE_{EC}$ drops again (-41.8W m$^{-2}$) followed by spike (149.5W m$^{-2}$) at 16:00 CET. For the two intense precipitation pulses at 13:00 and 16:00 CET, $LE_{EC}$ is flagged with 2 (bad data) according to the *ICOS* processing chain (Sabbatini et al., 2018). Hence, it can be assumed that rainfall with the intensity above a certain

threshold leads to issues in the spectral correction and thus unreasonable fluxes of $LE_{EC}$. Generally, Figure 4 indicates that high relative humidity, which coincides with precipitation and water stored on canopy, causes an underestimation of $LE_{EC}$ as shown by Massman and Ibrom (2008) or Zhang et al. (2023). While average $LE_{2D}$ is 26.8W m$^{-2}$ for the interception conditions after the second block of $P_g$ (17:00 to 24:00 CET), corresponding flux density of $LE_{EC}$ is only 2.8W m$^{-2}$ on average, despite addressing potential signal loss of water vapor fluxes due to tube attenuation or sensor separation in the data

processing (Fratini et al., 2012).

### 3.3    Adjustment of latent heat flux

#### 3.3.1    Conditions for implausible turbulent fluxes

The underestimation of $LE_{EC}$ can be quantified by the latent energy ratio ($LER$). $LER$ is the ratio of measured $LE_{EC}$ to $LE_{EB}$. With a closed energy balance, the residual is zero and $LE_{EC}$ equals $LE_{EB}$ ($AE - H$). Hence, $LER$ is one for a

closed energy balance and decreases with an increasing energy imbalance. Figure 5 a) shows a non-linear decreasing trend for the median of $LER$ along bins of increasing relative humidity $rH$ (periods with negative $LE_{EC}$ are not included in the analysis). The given values in panel a) show the number of half-hourly data points ($n$) for each bin of $rH$. Please note the





low amount of measurements for $rH$ 20% (n=22), which will be excluded in the following statistical analysis. The vertical lines for each point show the range of $LER$ from the $25^{th}$ to the $75^{th}$ percentile for each bin of $rH$. The median of $LER$ is between 0.55 and 0.62 for bins below 75% $rH$. For all bins exceeding 75% $rH$, $LER$ decreases markedly to a minimum of 0.17. This underlines a strong underestimation of $LE_{EC}$ for moist conditions with high relative humidity. Panel b) in Figure 5 highlights the dependency of high $rH$ and interception conditions. The curve also shows a non-linear dependency along bins of increasing relative humidity $rH$. The more water is stored on the canopy (increasing $C$) the higher $rH$. The median of $C$ starts to increase distinctively for bins exceeding 75% $rH$. Since only conditions with $C > 0$ are used in panel b), $n$ is a subset of the data used to calculate the $LER$ dependency on $rH$. Most of the data exceeding a relative humidity of 75% is measured under interception conditions. For example, more than 80% of the data exceeding 75% $rH$ in panel a) is represented in panel b), with an increasing median of $C$ from 0.5mm up to 8.7mm.

Since the $rH$ dependent underestimation of $LE_{EC}$ highly coincides with precipitation and interception, we substituted $LE_{EC}$ with $LE_{2D}$ for these conditions. Absolute and relative changes of $LE_{EC}$ after adjusting for interception conditions according to the modelled canopy water budget ($LE_{2D}$) are shown in panel c) and d) of Figure 5, respectively. Additionally, two correction methods based on the energy balance framework are shown for comparison: i) the Bowen ratio based energy balance closure for daytime conditions (global radiation $R_g > 20\mathrm{W\,m^{-2}}$) after Mauder (2013) $LE_\beta$ and ii) $LE_{EB}$ with all errors related to the energy imbalance (residual) attributed to the latent heat measurement. For the analysis, the same data source as in panel a) was used and for $LE_\beta$, data was additionally filtered for daytime conditions only, since no corrections are applied for nighttime conditions. This is also visible by the point size for each adjustment approach in panel c) and d), which is scaled according to the data size ($n$) for each bin of $rH$. The numbers above each point in panel c) and d) represent the median values for the respective approach. Quantifying the $LE$ adjustment for the residual approach $LE_{EB}$ also highlights the overall energy imbalance according to conditions of $rH$. In panel c), absolute changes of $LE_{EB}$, and hence the energy imbalance is highest for the smallest bins of $rH$ and the median steadily decreases from 71.39 to 1.98$\mathrm{W\,m^{-2}}$ until $rH$ 75%. From there, $\Delta LE$ is slightly increasing again for all bins up to $rH$ 95% from 3.04 to 20.16$\mathrm{W\,m^{-2}}$ for $LE_{EB}$. An over-closed energy balance, represented by a negative lower data quartile for $LE_{EB}$, occurs for bins of $rH$ between 40% and 75%. Absolute changes for the Bowen ratio adjustment $LE_\beta$ are highest at moderate $rH$ conditions with a peak at 40% (24.13$\mathrm{W\,m^{-2}}$) and a minimum at $rH$ 75% (9.89$\mathrm{W\,m^{-2}}$). Hence, absolute adjustments for $LE_\beta$ are most pronounced for dry situations with convective boundary layers. For conditions with the highest underestimation of $LE$ (sharp decrease of $LER$), absolute changes for the Bowen approach are almost constant, ranging between 8.24 and 11.85$\mathrm{W\,m^{-2}}$. In contrast, absolute changes for $LE_{2D}$ are only showing for bins of $rH$ exceeding 75%, since the adjustment is only applied for interception conditions. Absolute changes for $LE_{2D}$ show a sharp increase to a maximum of 13.72$\mathrm{W\,m^{-2}}$ at 85 and 90% $rH$, continuing to slightly decrease to 11.0$\mathrm{W\,m^{-2}}$ at 95% $rH$.

Relative changes in panel d) of Figure 5 show a different course along $rH$. The residual approach $LE_{EB}$ is showing a non-linear increase for bins of $rH$ exceeding 75%, which reflects the $rH$ dependency of $LER$ in panel a). The maximum relative change for $LE_{EB}$ occurs under the most moist conditions (95% $rH$) with a median of 416%. As expected, the adjustment of $LE_{EB}$ is the most drastic of all approaches. For the Bowen adjustment, the residual is partially attributed to $LE_\beta$, which





results in slowly decreasing relative changes for moderate $rH$ conditions up to 70% (median ranges between 19% and 36%). The $rH$ dependent error is only slightly accounted for with relative changes increasing from 41% to 92% for bins of $rH$ from 75% up to 95%. The adjustment incorporating the water budget for interception conditions reflects the $rH$ dependent error by a non-linear increase of relative changes for bins of $rH$ exceeding 80%. Substituted $LE_{EC}$ with $LE_{2D}$ for these interception conditions results in an increases from 21% up to a maximum of 284% at 95% $rH$.

### 3.3.2 Impact of LE adjustment on total evaporation

Absolute changes discussed in Figure 5 c) have shown that the Bowen ratio based energy balance adjustment for daytime conditions $LE_\beta$ accounts for the systematic error in $LE_{EC}$ associated with dry conditions. This method accounts for insufficient sampling of large-scale atmospheric motion, which is restricted to unstable stratification and prevalent in daytime convective boundary layers (Mauder, 2013). This concerns mostly moderate $rH$ conditions up to 75%, with no or low $rH$ dependency of $LER$. On the other hand, reliable water budget related estimates of evaporation $LE_{2D}$ can be used to correct low-pass filtering effects under conditions of high relative humidity, which affects especially closed-path systems such as *DE-THA* as shown in Figure 5 a). A high relative humidity highly coincides with precipitation and interception. This also includes stable or advective "wet-bulb" conditions, with a downward directed heat flux and suppressed turbulence. Consequently, we combined the two methods to arrive at a consistent dataset adjusted for dry and wet conditions. This new dataset $LE_{WB}$ incorporates the canopy water budget into the common practice to allocate the energy balance residual to the turbulent fluxes, in our case by preserving the Bowen ratio. First, $LE_{EC}$ was replaced by modelled data $LE_{2D}$ for interception conditions. The remaining "dry" dataset was then corrected on an half-hourly basis with the Bowen ratio based energy balance adjustment for daytime conditions after Mauder (2013).

The resulting $LER$ based on unadjusted sensible heat fluxes $H_{EC}$ is displayed in Figure 6 a) for the energy balance based correction methods $LE_\beta$ and $LE_{EB}$ and the combined water budget based adjustment $LE_{WB}$. The Bowen ratio conserving approach increases $LER$ to a quite constant value of 0.75 for low and moderate $rH$ conditions up to 70%. With increasing relative humidity $rH$, it follows the same non-linear decreasing trend as the uncorrected EC measurements. The residual approach yields full closure with $LER = 1$, since the energy imbalance is completely attributed to the latent heat flux. The combined approach $LE_{WB}$ leads to an increase of $LER$ along all bins of $rH$. As expected, the Bowen ratio and combined approach are similar for low and moderate $rH$ conditions up to 70%. For increased humidity conditions exceeding 70%, the decreasing trend of latent heat is removed for $LE_{WB}$ with $LER$ ranging between 89 and 65.

Figure 6 b) shows the average monthly course of total evaporation as water equivalent based on the $LE_{EC}$ measurements and the three half-hourly adjusted latent heat fluxes $LE_\beta$, $LE_{WB}$ and $LE_{EB}$. Average annual precipitation sum for the corresponding years is 1088±138mm. The annual water equivalent for available energy is 932±6mm. Hence, precipitation exceeds the available energy supply for potential evaporation. Monthly evaporation sums are smallest for the uncorrected EC measurement. Average annual sum of $LE_{EC}$ is 375±27.4mm, which accounts for 35±3% of annual precipitation. Evaporative fraction, which is the ratio of total evaporation to available energy ($E_{tot} : AE$) is 59±2%. The residual approach shows the highest monthly evaporation sums over all months of the year, except January and December, with an annual sum of 709±15.3mm.





Hence, attributing all energy balance residual to $LE_{EC}$ leads to an increase in evaporation that accounts for around $66\pm8\%$ of total precipitation and $76\pm2\%$ of available energy, respectively. The Bowen ratio based energy balance adjustment $LE_\beta$ shows the strongest effects from March till September, with the highest relative change of $LE_{EC}$ in Mai (37%), June (34%) and July 400 (38%). Average annual $E_{tot}$ is $493\pm10.5$mm, which accounts for $46\pm5\%$ of $P_g$ and $53\pm1\%$ of $AE$, respectively. Average sums of monthly evaporation for the combined water and energy budget based adjustment $LE_{WB}$ are located between the two other methods. Relative changes are highest from November to February, with a relative change of $LE_{EC}$ above 100%. However, $LE_{WB}$ exceeding the $LE_{EB}$ adjustments in January and December should be viewed critically, as this concerns months with snowfall, which is not treated separately from liquid precipitation by the 2D model. Average annual sum of $LE_{WB}$ is 405 $638\pm16.4$mm, which accounts for $59\pm6\%$ of $P_g$ and $68\pm1\%$ of $AE$, respectively.

## 4 Discussion

### 4.1 Comparing independent estimates of evaporation

A direct comparison of evaporation by different methods such as eddy covariance EC or canopy water balance WB is not so simple due to different source areas and uncertainties in the respective approaches. The evaluation of the results of both methods 410 presupposes on the one hand similar or homogeneous interception properties, such as meteorological conditions and vegetation characteristics, in the respective source areas. On the other hand, it is assumed that transpiration and evaporation from litter/soil are negligible for saturated conditions or sufficiently closed canopies. Then, measured total evaporation by the EC approach can be substituted by $E_I$. The application of the 2D Rutter model has shown that the classical canopy water balance approach is not statistically representative for the EC footprint area or a larger domain of the investigated spruce forest. The accuracy 415 of the WB approach depends on the structural characteristics of vegetation and on the precipitation regime. Spatially variable $T_f$ and $E_I$, as presented by the model results, require increased sampling efforts than the two throughfall collection gutters. Additionally, Zimmermann and Zimmermann (2014) reported generally higher relative throughfall sampling errors (up to 40%) during events with low intensities. Pluntke et al. (2023) suggested to use a scaling factor to compare the different source areas based on the LAI ratio of the average footprint area and the WB location. However, this is not in agreement with the 2D model 420 results for both areas. Modelled evaporation sums of the footprint area are about 0.9 times the amount of the WB location (regression not shown here), while the ratio of the respective PAIs is 0.71 (Pluntke et al. (2023) arrived at a ratio of 0.8).

The model results for the source area of the WB approach agree very well with the observations. However, some events caused by long ($> 10$hours) or intense precipitation ($> 10$mm h$^{-1}$) were underestimated by the model an could be further improved by an adjustment of storage parameters or drainage coefficients. Nevertheless, the 2D model was used confidently 425 as a reference for evaporation estimates under interception conditions. An earlier comparison with the EC measurements of latent heat by Fischer et al. (2023) already demonstrated a systematic underestimation of $LE_{EC}$, which was also concluded by Vorobevskii et al. (2022); van Dijk et al. (2015); Ringgaard et al. (2014). When compared for selected events, EC derived evaporation accounts for only 24% of total precipitation, which is an untypically low proportion for a dense coniferous forest under a temperate climate. Modelled total evaporation for the footprint area accounted on average for 54% of $P_g$ for the same



events. The analysis of an example event emphasized high deviations between the model reference $LE_{2D}$ and $LE_{EC}$ during precipitation and conditions with increased canopy water storage $C$. These conditions usually coincide with high relative humidity, which can lead to biases due to incorrect low-pass filtering of water vapor especially in closed-path systems (Zhang et al., 2023). More than 80% of the data exceeding a relative humidity of 75% can be attributed to an interception event. Interception conditions prevail on around 55±7% of all days of the year, of which 21±3% are with precipitation. Hence, a

majority of data is affected by the systematic underestimation effect of $LE_{EC}$ during interception. This applies also to the analyzed period 2008 to 2010, with an above-average annual precipitation sum of $1088\pm138\,\mathrm{mm\,a^{-1}}$ as compared to the long-term record for the period 1991-2020 with an average annual sum of $842\,\mathrm{mm\,a^{-1}}$.

## 4.2 Correction approach for EC measurements

The unaccounted energy as shown by $LER$, absolute and relative energy balance residual along bins of $rH$ can be mainly

explained by two different phenomena related to dry and wet conditions, respectively. In principle, the following explanation presupposes that systematic measurement errors are already minimized as much as possible in the data processing. Under dry or moderate $rH$ conditions up to 75%, no $LER$ dependency on humidity was detected. These conditions are generally associated with strong energy fluxes prevalent in daytime convective boundary layers under unstable stratification. Hence, the absolute potential underestimation of $LE_{EC}$ as shown by the differences to $LE_{EB}$ was highest under dry conditions. Bambach

et al. (2022) and Mauder et al. (2024) concluded, that the absolute median of the residual is greatest under very unstable and unstable conditions. The related systematic error can be explained by insufficient sampling of large-scale coherent eddies through secondary circulations. Assuming scalar similarity, the Bowen ratio of the measured fluxes is equal for large-scale structures, which allows to adjust the underestimated fluxes according to Mauder (2013). More recent adjustment methods to correct turbulent EC fluxes for secondary circulations such as proposed by Mauder et al. (2021) or Wanner et al. (2024) are

also applicable and should be considered for further studies.

For wet conditions with $rH$ exceeding 75%, a strong non-linear $LER$ dependency on humidity was detected. This $LE_{EC}$ underestimation effect highly coincides with precipitation and interception conditions when water is stored on the canopy. This concerns stable and advective "wet-bulb" conditions. Bambach et al. (2022) denote conditions of enhanced water vapor transport and low available energy as "pseudo-stable", since the stability caused by these circumstances does not follow the

classic definition of a generally suppressed turbulent transport. The study analyzed EC measurements of $LE$, EBC and nine correction methods for $LE_{EC}$ (residual and Bowen ratio based approaches) over irrigated vineyards, as well as uncertainties relative to atmospheric conditions. They found a generally larger uncertainty of $LE$ estimates across methods for days and sites with more prevalent daytime pseudo-stable conditions. Hence, solely energy balance based correction methods show strong deviations under various atmospheric conditions (Bambach et al., 2022), since the reasons for the underestimation of

fluxes differ. As a consequence, we extended the energy balance framework in the processing and interpretation of EC data by including the study site's water budget to obtain more reliable latent heat fluxes under wet or interception conditions. The validated 2D Rutter approach served as an independent estimate of $LE$ for interception conditions. Absolute and relative changes in $LE_{EC}$ were in agreement with the findings of Zhang et al. (2023). They corrected potential biases of $LE$ caused by





incorrect low-pass filtering of water vapor with a data driven machine learning approach. We suggest that a physically based
model is preferable to obtain reliable estimates of evaporation on a half-hourly time scale.

Consequently, we combined the two methods addressing different causes for the systematic underestimation of latent heat flux to arrive at a consistent dataset adjusted for dry and wet conditions. The new dataset $LE_{WB}$ yields an average annual increase of total evaporation of $263\pm26\mathrm{mm\ a^{-1}}$. Other than for the solely Bowen ratio based adjustment method, the new approach also shows a strong increase of evaporation for the winter half year, in which pseudo-stable conditions play a major
role. Vorobevskii et al. (2022) estimated evaporation for DE-Tha for the period 1997 to 2020 using the water budget model BROOK90 (Federer, 2002). They show that in particular the interception component for different model parameter sets is reduced after calibration to the EC flux data EC data adjusted for energy balance closure by a standard Bowen ratio preserving approach. Chapter 3 in Van Stan et al. (2020) states as a common result of several independent comparisons between models and EC measurements that modelled latent heat flux above forest is often overestimated and unlikely to match the (corrected)
EC measurement, while short vegetation and cropland is often reasonable well-modelled. A question which arises from these common observations is whether EC measurements of latent heat fluxes for forest ecosystems might be a reasonable reference for the calibration and validation of evaporation models, especially considering the systematically underestimated interception component and the still remaining decreasing relation between $LE_\beta$ along increasing $rH$.

Pluntke et al. (2023) estimated long-term total evaporation as the residual of the water balance for the Wernersbach catch-
ment, which is also located in the Tharandt Forest with a similar tree species distribution as the flux tower site DE-Tha. They retrieved an average annual $E_{tot}$ of $709\mathrm{mm\ a^{-1}}$ which corresponds to $77\%$ of precipitation for the period 2000 to 2009. This value is about $71\mathrm{mm\ a^{-1}}$ higher than our result after adjusting dry and wet conditions separately for quite wet years in 2008 to 2010. However, they explained that a difference between 40 and $85\mathrm{mm\ a^{-1}}$ could be due to different site conditions. They also compared their findings to the flux measurements of DE-Tha adjusted solely based on the energy balance and concluded, that
the differences between the two sites are too large to be explained by different site conditions alone. However, with independent measurements on transpiration and interception as well as roughly estimated soil evaporation and understory evaporation they estimated annual evaporation at DE-Tha and arrived at a value of 631 to $676\mathrm{mm\ a^{-1}}$ for 2006 to 2019, which is very close to our adjustment method $LE_{WB}$ with $638\mathrm{mm\ a^{-1}}$.

However, extending the energy balance framework by including the water budget in the processing and interpretation of EC
data requires either statistically representative throughfall measurements or a reliable canopy water budget model. Detailed information on the vegetation characteristics such as $PAD$ is necessary to match the footprint of the EC measurements and to spatially represent the investigated ecosystem. We demonstrated that our physically-based 2D Rutter model approach can reproduce sums of interception for the source area of independent canopy water budget measurements. Since the 2D model with a closed energy balance agrees very well with canopy water budget measurements, the model results of the EC footprint
confidently served as a reference to analyze and adjust EC-based evaporation. However, the model was only validated for liquid rainfall conditions and frost free periods, since throughfall measurements are only reliable during these conditions. The application of the results to the whole year, especially situations with snowfall, should be further investigated. Firstly, there is a lack of reference data and secondly, the modelling approach does not differentiate between solid and liquid precipitation. We



expect that our combined water and energy balance adjustment approach $LE_{WB}$ is still plausible, since snow interception for
DE-Tha is estimated less than 2% if distinguishing these processes (Vorobevskii et al., 2022). Additionally, a more accurate
representation of the precipitation and evaporation distribution could be made possible by considering the vegetation as a
volume in a 3D version of the model. The integration of a soil volume could account for vertical soil water movement and
storage. A further analysis of the model for a longer study period, also covering dry years or extreme precipitation conditions
would be interesting to test the performance of the model. Accurate process modelling will play an important role, given the
intensification of rainfall extremes and droughts (IPCC, 2021).

## 5   Summary and conclusion

Rainfall interception was analyzed from plot scale for the classical canopy water balance method (throughfall measurements)
to stand scale for the footprints of EC measurements. The results of a 2D modeling approach resembling a spatially variable
canopy structure were used for comparison and integration of the two measurement concepts. The study site - a typical managed
Norway spruce forest, located in a low mountain range close to Dresden, Germany - showed high amounts of interception evap-
oration $E_I$ accounting for about 52% of precipitation for all selected events and according to model output for approximately
50%. The study period 2008 to 2010 was characterized by relatively wet conditions, with an average annual precipitation of
1088±138mm. The 2D Rutter approach allowed a closer look at independent estimates of evaporation and the components
of the energy and water balance. The model was used to determine reliable estimates of interception evaporation for different
source areas. All quality measures showed very good agreement between the modelled and measured $T_f$ and $E_I$ for the 2D
approach. The results of a simple big leaf approach suffered from bias due to unaccounted spatially variable canopy structure,
which affects processes like precipitation partitioning and evaporation. Regardless of the chosen approach, all results revealed
a systematic underestimation of evaporation during selected events of interception by the EC method accounting for only 24%
of precipitation. We demonstrated that standard flux correction approaches are not appropriate for conditions of interception
and relative humidity exceeding 75% as also concluded by van Dijk et al. (2015) and Zhang et al. (2023).

As a consequence, we complemented field measurements with modelled estimates of evaporation to overcome the men-
tioned limitations and to arrive at a consistent dataset with adjusted latent heat fluxes for dry and wet conditions. We consider
this hybrid correction approach as a viable and pragmatic solution to adjust underestimated fluxes of $LE_{EC}$. This approach
considers on one hand unaccounted energy and on the other hand unaccounted water with the aim to close both budgets, which
are linked by evaporation. The sensible heat flux during rainfall interception is also affected by limitations of the measure-
ments during rainfall interception. Our investigations indicate significant downward directed sensible heat flux due to wet bulb
effects. However, this requires more research on boundary layer dynamics under stable and pseudo-stable conditions with
enhanced horizontal advection due to spatially variable wet and dry surface areas. Improved correction methods are urgently
needed, since EC systems are considered as the best available method to measure ecosystem-scale fluxes and for studying
global surface-atmosphere interactions. As EC data are typically used to parameterize *Soil Vegetation Atmosphere Transfer*



schemes and climate models as well as to calibrate remote sensing data, unaccounted interception in EC measurements would lead to a systematic bias in all kinds of applications.

Forest ecosystems, particularly dense evergreen forests appear to be affected by the underestimation of heat fluxes during interception due to their large capacity for water and energy storage. The impact of this effect needs to be further investigated for different altitudes and ecosystem types. Accurate estimates of interception play an important role in assessing the water availability in ecosystems in order to maintain their growth and function. Additionally, the amount of water captured by canopies has an impact on the rainfall-runoff distribution. Lian et al. (2022) globally analyzed the impact of altered precipitation on interception and detected a decreasing trend due to reduced rainfall partitioning with less frequent and more intense rain events. Shifts in rainfall and interception characteristics and their respective response to ecosystem water availability, erosion or flood risks require further investigation. This in turn requires reliable data sets and procedures to systematically identify and quantify the sources for unaccounted energy and water, which depend on atmospheric conditions, ecosystem and site characteristics.

## Appendix A:  Model of total evaporation

### Model structure

The primary objective of the model development is to estimate the total evaporation of both, transpiration and interception in the footprint area of the eddy covariance (*EC*) measurements, also during system failures. This estimation is crucial because the footprint area changes with wind direction and atmospheric conditions, which in turn changes the vegetation cover and the amount of evaporation from intercepted rainwater that is covered by the *EC*-system, while practically all existing setups to derive interception from the difference of gross and canopy precipitation will cover only a small sample in a certain direction relative to the flux tower (here a troughfall measuring system). Therefore, it is essential that the model accurately represents this horizontal variability. The effectiveness of the model can be evaluated using the measured water balance of the trough system.

To address these challenges, a horizontally structured *big leaf* approach is proposed, utilizing grids of 10m x 10m. For each grid, the water balance equation is solved following the conceptual framework established by Rutter et al. (1971). The model incorporates several key equations that govern precipitation partitioning, storage, drainage, evaporation and transpiration, all of which are influenced by the Plant Area Index PAI. This method considers horizontal variability in the vegetation stand and the calculation of transpiration and interception evaporation weighted by the footprint distribution of the *EC*-measurements.

Equations A1 and A2 contain the components of the estimated total evaporation $E_{tot,est}$ that should be covered by the *EC* measurement system. $E_T$ and $E_I$ are transpiration and evaporation of the vegetation, respectively. The evaporation of the intercepted rainwater is a very dynamic process which is why Equation A2 is written in intensities of evaporation $E_I$, precipitation $P_g$, free throughfall $T_f$, stemflow $T_s$, drainage $T_d$ and the storage change $dC/dt$ .





$$E_{tot,est} = E_T + E_I \tag{A1}$$

$$\dot{E}_I = \dot{P}_g - \dot{T}_f - \dot{T}_s - \dot{T}_D - \frac{dC}{dt} \tag{A2}$$

The equations for these components are given in detail in the sub-sections that follow. The interception model has been
implemented in R®, facilitating easy and interactive use for users.

**Precipitation partitioning**

In dependence of the $PAD$, precipitation is partitioned into three main components:

(1) Interception and storage $C$: Water that is intercepted by the canopy and subsequently evaporates.

(2) Throughfall $T_f$: Rainfall that passes through gaps in the canopy.

(3) Stemflow $T_S$: Water that flows down the stems of plants

Stem flow is considered negligible and was not regarded for the Norway spruce stand under investigation. Throughfall is
calculated as the part of the gross precipitation that is falling freely through canopy gaps $p_{T_f}$. Interception of rainwater filling
the canopy storage $C^+$, is the remaining part of the precipitation (Eq. A3 and A4).

$$T_f = P_g \cdot p_{T_f} \tag{A3}$$

$$C^+ = P_g - T_f \tag{A4}$$

$p_{T_f}$ is described by Equation A5, in which canopy closure with increasing $PAI$ is taken into account by a smoothing
function.

$$p_{T_f} = \begin{cases} 0, & \text{if } PAI > cc_2 \\ 1 - \frac{1 - e^{(-PAI\frac{cc_1}{cc_2})}}{1 - e^{(-cc_1)}}, & \text{otherwise} \end{cases} \tag{A5}$$

Equation A5 is controlled by the parameters $cc_1 = 2.5$ responsible for the convexity of the function and $cc_2 = 15\mathrm{m}^2/\mathrm{m}^2$
that corresponds to the PAI when the canopy is fully closed (please note that $cc_2$ may be subject to change depending on the
grid size).

Figure A1 demonstrates different shapes of the function.



**Drainage from interception storage**

With exceeding canopy storage capacity, excess water drains off the canopy with increasing speed. Rutter et al. (1971) proposed

an exponential increase in drainage that is described by Equation A6. This process is influenced not only by the Plant Area Index but also by external factors such as rainfall intensity and wind speed. In the approach of Rutter et al. (1971) only the PAI is regarded implicitly within the canopy storage capacity S, which leads to additional scatter comparing the model results with independent throughfall measurements.

$$T_D = D_{min} \cdot e^{b \cdot (C-S)} \tag{A6}$$

The storage capacity $S_0$ is determined from measurements taken for a certain $PAI_{trough}$ representing the throughfall measuring system. Since this study involves variable PAI values, $S_0$ is adjusted for each grid cell using Equation A7 to reflect the specific PAI in the respective cell.

$$S = S_0 \frac{PAI}{PAI_{trough}} \tag{A7}$$

The drainage coefficient was determined at $b = 3.7 \text{mm}^{-1}$ from Rutter et al. (1975), based on measurements in a Corsican

Pine stand with the storage capacity of $S_{CP} = 1.05 \text{mm}$. Additionally, the minimal drainage parameter was set to $D_{min,CP} = 0.002 \text{mm min}^{-1}$ and is defined as minimal drainage rate. To adjust the value for this study, this value was scaled with the storage capacities of both stands, $D_{min} = D_{min,CP} S_0/S_{CP}$.

**Total Evaporation**

The Penman-Monteith equation is a widely used model that combines flux-gradient relationships with the energy balance equa-

tion to estimate total evaporation. This approach integrates the "loss" of latent heat through both evaporation and transpiration (Monteith and Unsworth, 2008). Thus, both components are treated as a combined process in the following model. In the Equation A8, the latent heat flux is converted into its water equivalent, denoted as $ET_{mod}$, using the latent heat of vaporization. This conversion allows the model to express total evaporation in terms of water loss. Key variables in the Penman-Monteith equation include: the radiation balance $Rn$, the water vapor pressure in air $e$, the saturation vapor pressure $e_s(T)$ at temperature

$T$, change of $e_s$ with temperature $\Delta = de_s/dT$, the psychrometric constant $\gamma$, the density of air $\rho_a$ and the heat capacity of air $c_p$.

    The resistances against the transport of latent heat and sensible heat are denoted as $r_{LE}$ and $r_H$, respectively. Whereas $r_{LE}$ combines the transport from a wet canopy surface $r_{c,w}$ (i.e. evaporation) and the transpiration through the stomata $r_{c,s}$. Both paths operate in parallel. The parameter $sc$, the saturated part of storage capacity, regulates which path is preferred (Eq.

A9). The transport resistance from a wet canopy surface takes the quasi laminar boundary resistance $r_b$ and the turbulent boundary resistance $r_a$ into account, $r_{c,w} = r_b + r_a$, whereas the $r_{c,s}$ additionally includes the canopy stomatal resistance $r_s$,



i.e. $r_{c,s} = r_s + r_b + r_a$. The three resistances are determined according to approaches of Stewart (1988) and Jarvis P. G. et al. (1976) for $r_s$, Jensen and Hummelshøj (1995) for $r_b$ and Monteith and Unsworth (2008) for $r_a$.

$$E_{tot,est} = \frac{\Delta Rn + \varrho_a c_p \left(e_s\left(T\right) - e\right)/r_H}{L_v(\Delta + \gamma(1 + \frac{r_{LE}}{r_H}))} \tag{A8}$$

$$r_{LE} = \left(\frac{1 - sc}{r_{c,s}} + \frac{sc}{r_{c,w}}\right)^{-1} \tag{A9}$$

As an initial estimate, $sc$ could be represented by the relative filling of the canopy storage, defined as $sc_0 = C/S$. However, this approach does not take into account that the saturated layer, which is replenished by the droplets on the surface, accumulates rapidly with the first few drops and then more gradually thereafter. Equation A10 addresses this phenomenon and functional shapes are illustrated in Figure A2.

$$sc = \begin{cases} \frac{1 - e^{-((1 - sc_{min}) * sc_0 + sc_{min}) * sc_f}}{1 - e^{-sc_f}}, & \text{if } C < S \\ 1, & \text{otherwise} \end{cases} \tag{A10}$$

The partitioning of total evaporation $E_{tot,est}$ into evaporation and transpiration can be achieved by using the ratio of the individual resistances to the total resistance against the latent heat flux (Eq. A11 and A12).

$$E_{I,mod} = E_{tot,est} \frac{r_{LE}}{r_{c,w}} \tag{A11}$$

$$T_{mod} = E_{tot,est} \frac{r_{LE}}{r_{c,s}} \tag{A12}$$

*Data availability.* Flux and meteorology data for the study site DE-Tha is available on various platforms such as FLUXNET and ICOS.

*Author contributions.* The study was conceptualized by SF, RQ, CB and MM. Data preparation, analysis, visualization and the preparation of the original manuscript was done by SF. RQ developed the 2D Rutter model and prepared the appendix. Model configuration was conducted by SF and RQ. RQ, CB and MM supervised the study, reviewed and contributed to the manuscript.

*Competing interests.* The authors declare that they have no competing interests.



*Acknowledgements.* This work was supported by the German Science Foundation DFG BE-1721/23-1. We would like to thank the site manager of the ICOS-D ecosystem cluster including the Anchor Station Tharandt Thomas Grünwald for his valuable feedback on the data basis. We also acknowledge the support of Uta Moderow, the technical assistance of Heiko Prasse and Markus Hehn, as well as the scientific discussions with all colleagues of the Chair of Meteorology, TU Dresden. Finally yet importantly, we gratefully acknowledge the valuable comments and suggestions of the reviewers.



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



**Figure 1.** Vertical integrated plant area density in $\mathrm{m^2\ m^{-2}}$ ($PAI_{Local}$) of the model domain of the study area derived from terrestrial laser scanning (resolution $1\mathrm{m^2}$). The triangle indicates the EC tower and the v the two gutters for throughfall measurements. The map at the top left shows the location of the study area in the Tharandt Forest (©OpenStreetMap contributors 2024. Distributed under the Open Data Commons Open Database License (ODbL) v1.0.)





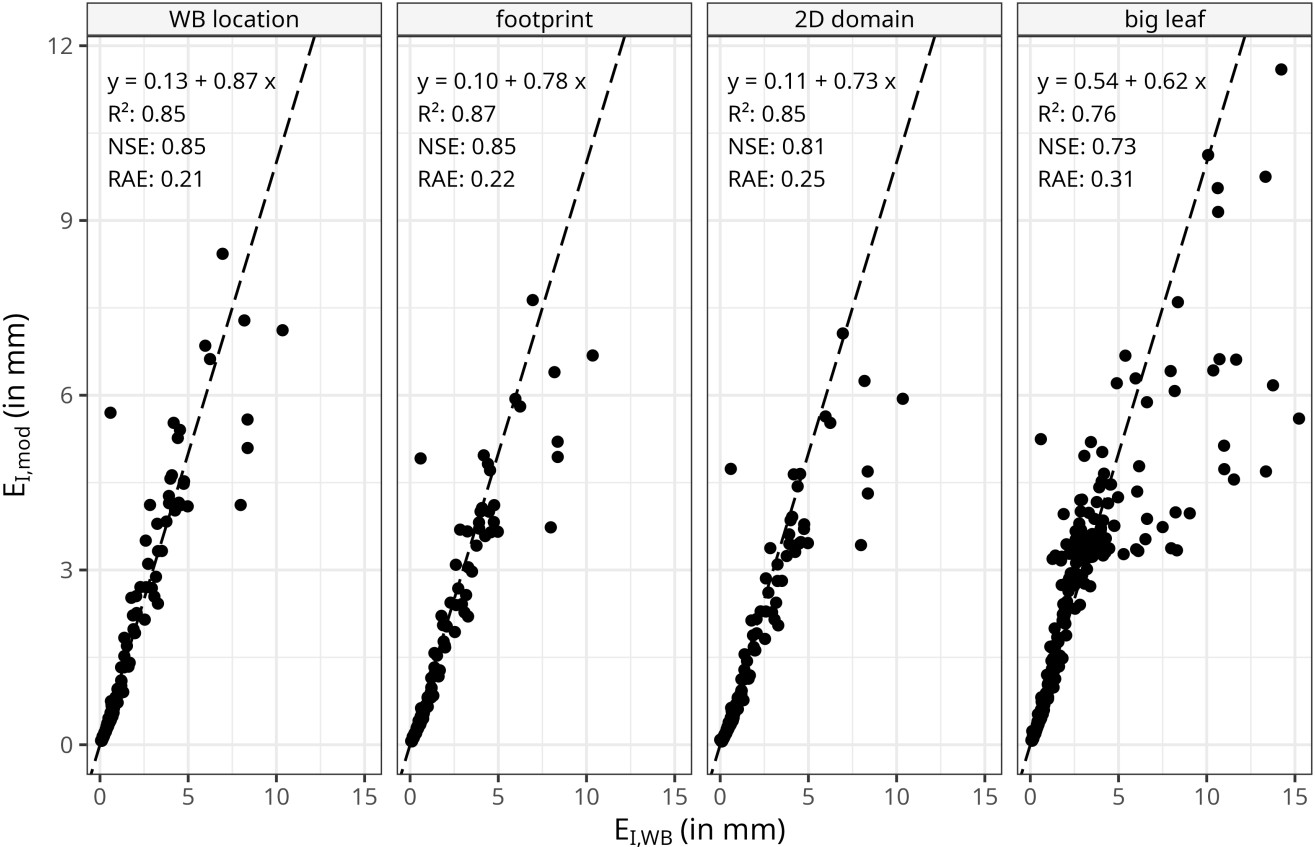

**Figure 2.** Event based interception evaporation ($E_{I,mod}$) of the 2D model (n=149) for different source areas: i) water balance WB approach (WB location), ii) footprint area of the EC measurements (footprint), iii) whole model domain (2D domain) and for a simple big leaf approach (n=302). Interception measurements $E_{I,WB}$ (retrieved from collected gross precipitation and throughfall) in the vicinity of the EC measurement system serve as reference.





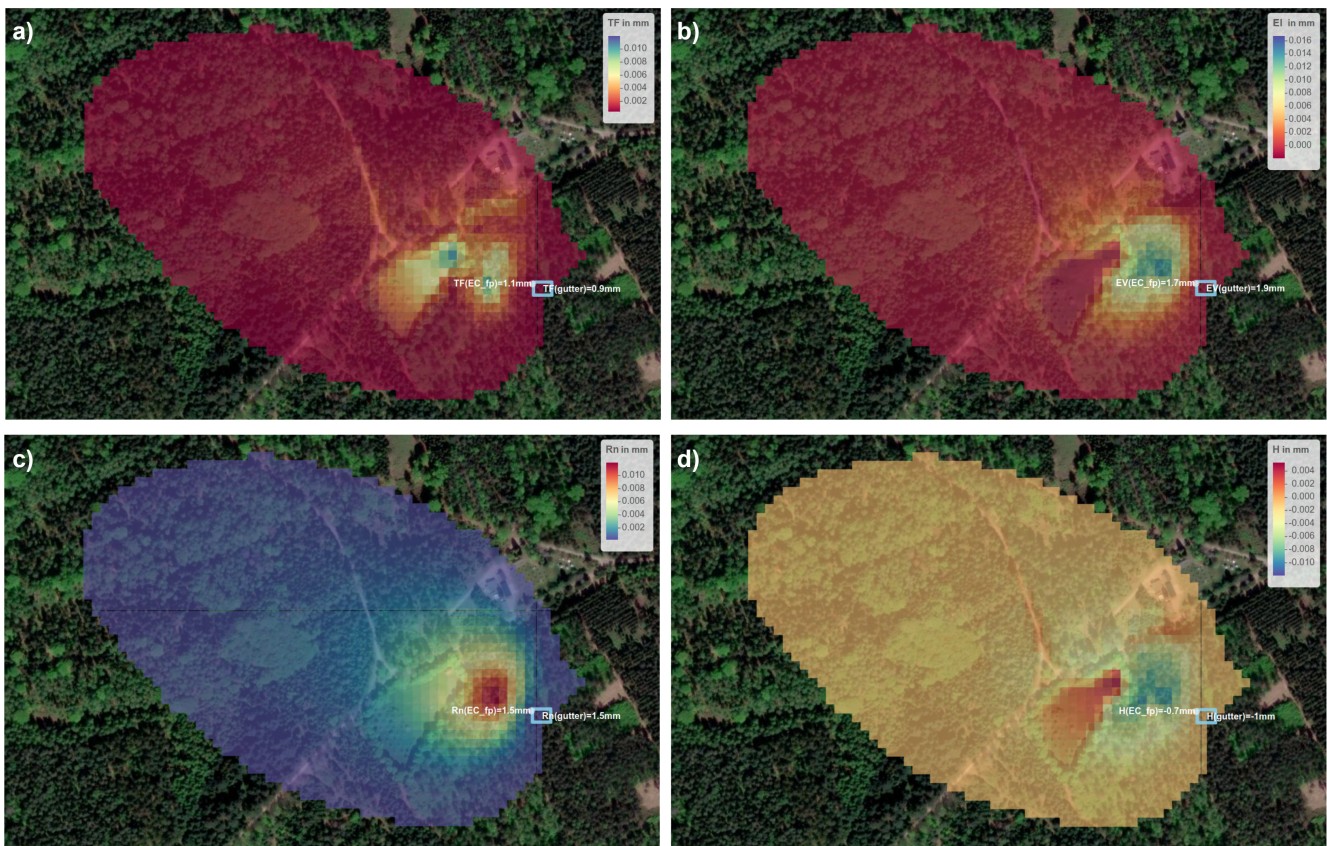

**Figure 3.** Spatial distribution of water and energy budget related components in mm for a temporally aggregated interception event of 24.5h duration (start fourth October 2009 11:40 CET till fifth October 2009 12:10 CET). Spatially averaged values are shown for the flux footprint area ($EC\_fp$) and the throughfall plot (*gutter*). The panels show the EC related flux footprint and modelled thoughfall $T_f$ a), interception evaporation $EI$ b), associated net radiation $Rn$ c) and sensible heat flux $H$ d). The map was created using the R leaflet package (source: Esri, i-cubed, USDA, USGS, AEX, GeoEye, Getmapping, Aerogrid, IGN, IGP, UPR-EGP, and the GIS User Community).



**Figure 4.** Meteorologic drivers, water and energy budget related components for the onset of an interception event at 12:00 CET at fourth of October 2009 (same as in Figure 3). The panels show (from top to bottom) precipitation $P_g$, modelled canopy water storage $C$, relative humidity $rH$, horizontal wind velocity $u$, net radiation $Rn$ and turbulent fluxes $LE$ and $H$ in half hourly resolution. Turbulent fluxes are shown as measured by the EC system (solid line) and modelled by the 2D Rutter approach (dot-dashed line).

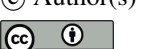



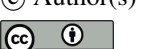

**Figure 5.** Underestimation of half-hourly $LE_{EC}$ and its dependency on relative humidity $rH$ quantified by latent energy ratio $LER$ a). A relationship of contributing half-hourly interception conditions with a canopy water storage $C > 0$ and relative humidity is shown in panel b). The numbers above the points in a) and b) are the total amount $n$ of half-hourly data points and vertical lines are the inter-quartile ranges. Panel c) and d) show absolute and relative changes in $LE_{EC}$ across bins of $rH$ depending on the method used to estimate $LE$: i) Bowen ratio based for daytime conditions $LE_\beta$ (only half-hourly data for daytime conditions shown), ii) canopy water budget related estimates $LE_{2D}$ for interception conditions and iii) energy balance residual attributed to the latent heat $LE_{EB}$. The numbers above the points in c) and d) are the median value and vertical lines are the inter-quartile ranges.





**Figure 6.** Latent heat flux data based on EC measurements $LE_{EC}$ and three different adjustment methods: i) Bowen ratio preserving for daytime conditions $LE_{\beta}$, ii) combination of canopy water budget related estimates for interception conditions and Bowen ratio based (as for the previous) for the remaining dry conditions $LE_{WB}$ and iii) energy balance residual attributed to the latent heat $LE_{EB}$. Panel a) shows the latent energy ratio $LER$ for all data sets based on unadjusted sensible heat flux ($LE_x/(AE - H_{EC})$), similar to Figure 5 a). Panel b) displays mean monthly total evaporation $E_{tot}$ for the years 2008 to 2010 as water equivalent in mm. Annual variability is highlighted for each method as the range of standard deviation. Mean annual precipitation $P_g$ for 2008 to 2010 is 1088±138mm.



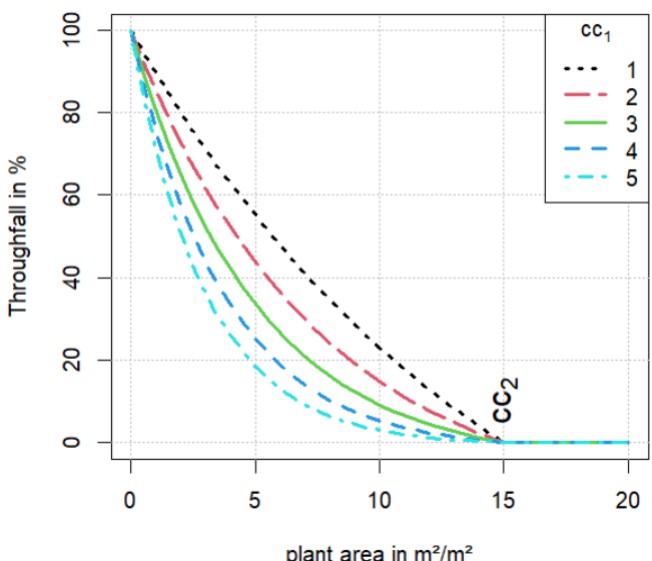

**Figure A1.** Free throughfall coefficient: part of the gross precipitation that is falling freely through canopy gaps

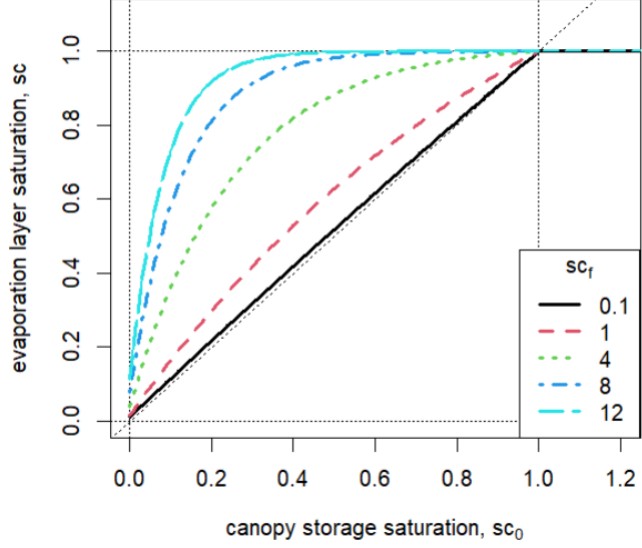

**Figure A2.** Estimated development of the relative saturated plant surface area $sc$ as a function of storage and storage capacity $sc_0$.