# Peer review of "Quantifying evaporation of intercepted rainfall: a hybrid correction approach for eddy-covariance measurements"

_EGUsphere, 2025_

## Referee Comment (RC1)

**Quantifying evaporation of intercepted rainfall: a hybrid correction approach for eddy-covariance measurements**
by Stefanie Fischer, Ronald Queck, Christian Bernhofer, and Matthias Mauder

Reviewed by: David R. Fitzjarrald, Atmospheric Sciences Research Center, UAlbany, SUNY, US of A

**General comments**

This is a well-researched paper. The authors' diligent work should be recognized and the work published. Let there be sufficient revisions that my questions and comments are addressed.

**Introductory distraction:**

Thirty-one years ago, during the BOREAS project, our team measuring eddy fluxes in a jack pine stand in northern Manitoba, Canada had a trough system set up to attempt to measure throughfall, but there was so little rain that nothing came of this effort, save for an anecdote. In frustration, one evening I reported during the daily project conference call that we had installed a second rain gauge *upside down,* so as to measure *incident and reflected rainfall.* The Program Manager is said to have dutifully put this information into the notes until noticing that everyone had collapsed in hilarity. OK, but is not the return of intercepted water indeed a kind of 'reflected' precipitation?

**Overview of the problem at hand and its attendant observational difficulties:**

General aspects of the problem if estimating rainfall interception in forests: Observations presented in this paper include direct, inferred, and some hybrid of observations & modeling. If the water returned to the atmosphere as vapor is to be inferred from perturbations in energy balance as obtained by eddy covariance measurement, how are these to be made believable?

Fischer et al. here manipulate the *processed standardized* Fluxnet data in new ways, complementing this with model output, apparently in the hopes that the rainfall interception estimate is "improved", in the sense that the results reported in their earlier article (Fischer et al., 2023) might more closely resemble those published at similar Fluxnet forest sites. The manner in which these flux data may already have been manipulated by the Fluxnet hierarchy is noted early in the text: (manuscript Line 42: "the eddy covariance data itself require complex setups, data handling (esp. post processing) and careful interpretation. …".

The authors emphasize that the difficulty of achieving "energy balance closure" was *the* fundamental question about using eddy flux data to assess the *return* of intercepted rainfall to the lower atmosphere as vapor. There are also additional issues that can hinder adequate accounting of the transient period of re-evaporation after rainfall interception. The authors owe it to their readers to address these possibilities, and, when possible, make a vigorous defense of their approach. What is the fraction of 'flagged' or corrected data during and after the precipitation event? That is, how many of their conclusions is based on 'data inference'?

To be fair, with widely spaced comments in the body of this text and in the 2023 paper, the authors *do* note that during precipitation there are periods when data is suspect: Line 315: "For the two intense precipitation pulses at 13:00 and 16:00 CET, LEEC is flagged with 2 (bad data) according to the ICOS processing chain (Sabbatini et al., 2018)…." They did not make clear to me what fraction of

their data are of adequate eddy flux quality during low-wind conditions, a situation known to present such a quandary to the eddy covariance industry that local empirical models (like the so-called $u_*$ correction) are regularly inserted into what is then referred to as 'data'. I bring this up because this paper (Section 4.2) makes much of the issue of the credibility of interception estimates by flux during during high relative humidity conditions, but are these not very often *low-wind speed conditions,* for which the requirement of 'continuous turbulence', dear to Vickers and Mahrt (1997), is essential to forming the eddy covariance ensemble, may not be satisfied?

[From Fischer et al. (2023): "Data records of weak variance, potentially occurring under stable conditions or with low wind speeds have been detected after the approach of Vickers and Mahrt (1997)."

"1) The systematic error is due to the failure to capture all of the largest transporting scales, typically leading to an underestimation of the flux.

**2) The random error is due to an inadequate sample of the main transporting eddies as a consequence of inadequate record length.**

**3) The mesoscale variability or inhomogeneity (non-stationarity) of the flow can lead to a significant dependence of the flux on the choice of averaging scale."** Vickers and Mahrt (1997)

During near-calm conditions or with intermittent mixing, eddy fluxes as suspect. If one *could* average over longer periods of time, perhaps this would be overcome, but given that the objective here a is to deal with a *transient* event, this is not possible. See, for example, Medeiros & Fitzjarrald (2015).

***Some general questions for the authors to reply to:***

1. *How do the authors verify that current interception estimates are inadequate?*
2. *How are known issues with the eddy flux approach during rainfall addressed in this standardization?*
3. *How many actual observations (not "gap-filled") are in these Fluxnet evaporation data series during and after rainfall?*
4. *The authors state that intercepted rain is returned to the atmosphere 'immediately after' the rainfall, but this is not the case with nocturnal rain, for which intercepted water may not clear the canopy until the follows day's convective surface layer re-establishes. Czikowsky and Fitzjarrald (2009) found that rainfall intercepted during nocturnal rainfall evaporated during the first convective hours of the following morning. How about that? Please comment how this is dealt with. This has to do with how one decides what part of the subsequent LE 'belongs' the overnight rainfall.*
5. *Can the authors address the issue of interception as a function of rainfall rate?*
6. *Do the authors only use the Vaisala RH sensor to determine this quantity? Is there evidence that it reports as accurately as promised by the manufacturer near saturation conditions? Many such instruments do not do well then, and sometimes manufacturers overlook this issue, especially when spec sheets are made. Over what RH range do you trust the sensor? Can it compared with observations from other instruments on site?*

**—> Some detailed comments/questions are embedded in the *pdf* text file, attached.**

from the earlier paper, Fischer et al. (2023):

"The majority of rainfall events (81%) is characterized by a depth <5 mm, which "leads to a high fraction of annual precipitation being captured by the canopy surface (0.41). The application of the Rutter model yielded good agreement between observed and modeled throughfall and served as a reasonable standard to define interception events."

"The mean annual underestimation of $E_{mass}$ is 145 mm/year for $E_{TEB}$ and 288 mm/year for $E_{TEC}$. Most of the discrepancy can be mainly explained by an underestimation of turbulent fluxes, at which data is most affected during raining conditions. An analysis of the linear relation between the annual sum of turbulent fluxes and available energy shows the lowest slope (0.57 ± 0.15) for measurements during rainfall, while the highest slope (0.76 ± 0.03) occurs under completely dry conditions. Both fluxes should be gap-filled and corrected separately for dry (transpiration) and wet (interception) conditions in order to determine proper amounts of evapotranspiration with the eddy covariance method."

"… *the annual sum of turbulent fluxes and available energy*… Both fluxes should be gap-filled and corrected separately for dry (transpiration) and wet (interception) conditions in order **to determine proper** amounts of evapotranspiration with the eddy covariance method."

A little history & some relevant anecdotes:   Near a pond about 10 km by road from where I write this review, Horton (1919), made a series of direct observations of rainfall interception. Adding his results to earlier findings, this paper examined the percentage of interception, emphasizing the importance of species type, season, and relevant to the current paper, **rainfall intensity**.

[Figure]

FIG. 13.— Mean curve showing total per cent of precipitation in a shower intercepted by various trees in 1917-18.

Direct observation: *Plastic sheets on the rain forest floor to get a more complete throughfall observation!* Forty-one years ago, the Shuttleworth/UK ABRACOS project at the tower in Ducke Reserve near Manaus Amazonas, Brazil (Shuttleworth *et al.*, 1988) were making interception estimates in the rainforest. The UK/Brazil team applied lessons from the Institute of Hydrology to the tropical case. "The presence of such large sampling errors in the throughfall data for this forest means that comparison between experimental measurements and model estimates of the interception component is only useful in the form of cumulative totals, with many gauges frequently and randomly moved over a long period (Lloyd et al. I988). "The present experiment measured the integrated, fractional interception loss over the complete study period as 9.0 + 3.5 % of gross rainfall."

[Figure]

LBA-ECO.  *Event-based ensembles were found to estimate the eddy fluxes LE and H above a forest in Brazil. The basic eddy covariance data was recalculated in such a way that measurements were filtered to continue returning values during rainfall, and, re-starting the calculation at the moment the rain stopped, flux calculation periods were begun.*  Consider two "treatments" over the same forest, first a base state ensemble: days without rain, under the same radiative conditions as rain days. The second group is a precipitation event ensemble. This approach works well in the regular weather patterns of the tropics; perhaps it would be problematic in Europe.  It is incorrect to state the this approach has no error estimate, as has appeared in the literature.

*7. Do you think a similar approach could work at the Norway spruce conifer forest in Europe considered in this paper?*

[Figure]

Left: Identifying re-evaporation of intercepted rain from deviation from an ensemble average; Right: various estimations of the rainfall interception percentage in rainforests. ("15B" case of  Czikowsky & Fitzjarrald, 2009)

References.

Czikowsky, M.J. and Fitzjarrald, D.R., 2009. Detecting rainfall interception in an Amazonian rain forest with eddy flux measurements. *Journal of Hydrology*, *377*(1-2), pp. 92-105.

Fischer, S., Moderow, U., Queck, R. and Bernhofer, C., 2023. Evaporation of intercepted rainfall–Comparing canopy water budget and energy balance related long term measurements at a Norway spruce site. *Agricultural and Forest Meteorology*, *341*, p. 109637.

Horton, R.E., 1919. Rainfall interception. *Monthly weather review*, *47*(9), pp. 603-623.

Lloyd, C.R., Gash, J.H. and Shuttleworth, W.J., 1988. The measurement and modelling of rainfall interception by Amazonian rain forest. *Agricultural and Forest Meteorology*, *43*(3-4), pp. 277-294

Medeiros, L.E. and Fitzjarrald, D.R., 2015. Stable boundary layer in complex terrain. Part II: Geometrical and sheltering effects on mixing. *Journal of Applied Meteorology and Climatology*, *54*(1), pp. 170-188.

[revised manuscript text omitted]

---

## Author Comment (AC1)

Thank you very much for the constructive review of our manuscript, also for appreciating this work and supporting to improve it by valuable remarks and suggestions within the manuscript. We carefully went through all the comments and revised the manuscript accordingly. Replies to comments relating to specific lines, paragraphs or tables have been incorporated accordingly. Since the upload of the revised manuscript is not possible during this stage of publishing, revised sections are given by page and line number. Additionally, we address specific questions and major points of the reviewer below.

**Reviewer #1:**

Overview of the problem at hand and its attendant observational difficulties:

General aspects of the problem if estimating rainfall interception in forests: Observations presented in this paper include direct, inferred, and some hybrid of observations & modeling. If the water returned to the atmosphere as vapor is to be inferred from perturbations in energy balance as obtained by eddy covariance measurement, how are these to be made believable? Fischer et al. here manipulate the processed standardized Fluxnet data in new ways, complementing this with model output, apparently in the hopes that the rainfall interception estimate is "improved", in the sense that the results reported in their earlier article (Fischer et al., 2023) might more closely resemble those published at similar Fluxnet forest sites. The manner in which these flux data may already have been manipulated by the Fluxnet hierarchy is noted early in the text: (manuscript Line 42: "the eddy covariance data itself require complex setups, data handling (esp. post processing) and careful interpretation. ...".

The authors emphasize that the difficulty of achieving "energy balance closure" was the fundamental question about using eddy flux data to assess the return of intercepted rainfall to the lower atmosphere as vapor. There are also additional issues that can hinder adequate accounting of the transient period of re-evaporation after rainfall interception. The authors owe it to their readers to address these possibilities, and, when possible, make a vigorous defense of their approach. What is the fraction of 'flagged' or corrected data during and after the precipitation event? That is, how many of their conclusions is based on 'data inference'?

To be fair, with widely spaced comments in the body of this text and in the 2023 paper, the authors do note that during precipitation there are periods when data is suspect: Line 315: "For the two intense precipitation pulses at 13:00 and 16:00 CET, LEEC is flagged with 2 (bad data) according to the ICOS processing chain (Sabbatini et al., 2018)...." They did not make clear to me what fraction of their data are of adequate eddy flux quality during low-wind conditions, a situation known to present such a quandary to the eddy covariance industry that local empirical models (like the so-called u\* correction) are regularly inserted into what is then referred to as 'data'. I bring this up because this paper (Section 4.2) makes much of the issue of the credibility of interception estimates by flux during during high relative humidity conditions, but are these not very often low-wind speed conditions, for which the requirement of 'continuous turbulence', dear to Vickers and Mahrt (1997), is essential to forming the eddy covariance ensemble, may not be satisfied?

[From Fischer et al. (2023): "Data records of weak variance, potentially occurring under stable conditions or with low wind speeds have been detected after the approach of Vickers and Mahrt (1997)."

- "1) The systematic error is due to the failure to capture all of the largest transporting scales, typically leading to an underestimation of the flux.
- 2) The random error is due to an inadequate sample of the main transporting eddies as a consequence of inadequate record length.
- 3) The mesoscale variability or inhomogeneity (non-stationarity) of the flow can lead to a significant dependence of the flux on the choice of averaging scale." Vickers and Mahrt (1997) During near-calm conditions or with intermittent mixing, eddy fluxes as suspect. If one could average over longer periods of time, perhaps this would be overcome, but given that the objective here a is to deal with a transient event, this is not possible. See, for example, Medeiros & Fitzjarrald (2015).

**Page 2: line 55**

We restructured the statement to:

"EC measurements during stable and calm conditions poorly cover vertical transport and are questionable, since the role of storage and advection terms, such as the "wetbulb effect" (horizontal advection from dry areas), as well as insufficient sampling of low-frequency and large-scale motions remains unknown (van Dijk et al., 2015; Stoy et al., 2019; Fischer et al., 2023)."

The issue of time of the day of precipitation on the subsequent re-evaporatin of intercepted rainfall is addressed by the model approach, in which the water and energy budget of the horizontally variable canopy are calculated dynamically. Please see Figure 1 below.

**Page 5: line 131-134:**

We restructured the statement to:

"Forest floor interception  $E_{I,FF}$  is difficult to measure as it is very heterogeneous on the small scale. Gerrits et al. (2010) summarize the importance of  $E_{I,FF}$ , which is nearly constant throughout the year and accounts on average for 22% of throughfall. However, evaporation of intercepted water from the litter takes longer than that from the canopy, thus we assume that the forest floor evaporation during an event is negligible. Alternatively, it could be be addressed by additional measurements combined with models, which is beyond the scope of the study."

If forest floor interception would be accounted for, the systematic underestimation of  $E_{tot,EC}$  would be even higher. We are neglecting forest floor evaporation in the model as well, to compare both methods (2D Model and WB approach).

**Page 6: line 157-158**

From Fischer et al. (2023): "A Bowen-ratio-preserving correction during events of interception might not be reasonable since available energy close to zero or opposite signs but the same magnitude for H and LE lead to unreasonable results. These (stable) conditions are likely to occur for rain or interception events with cloudy conditions and a sink of sensible heat (lateral advection). Unfortunately, this approach often leads to dubious fluxes for the case of precipitation or interception (Bowen-ratio  $\approx$  1), when available energy AE is low or when H becomes a source of energy. Until now, no common agreement was reached on a solution for that specific situation." Thus, the approach to attribute the systematic error in the EC flux measurements entirely to latent heat LE to close the energy balance was taken as an upper limit for estimated latent heat fluxes. Additionally, LEEB is used to quantify the latent energy ratio LER (ratio of measured LEEC to LEEB).

**Page 7: title 3.1:**

We have changed the title of the paragraph to "Model evaluation".

**Page 9: line 256**

What kind of information about a footprint source area would you find for stable, nocturnal conditions?

The model of Kljun et al. (2015) requires parameters such mean wind speed, standard deviations of vertical wind, friction velocity, Obukhov length, displacement height, roughness length etc. to compute the footprints. It is applied for day and night time conditions including moderately stable conditions.

Model assumptions are turbulent stationary flow, continuous mixing and the validity of the Monin-Obukhov similarity theory. Under strongly stable conditions, turbulence is weak with a low friction velocity. As a result, predicted footprints may be unreliable and unrealistically large, displaced or physically meaningless. In our study, we focused on events of interception, which can also occur during night time but for which atmospheric conditions are not necessarily strongly stable and the model output was reliable. In some cases we excluded footprints of very high distances, since we evaluated only events with a footprint coverage inside the extend of our study domain.

"Modelled events were filtered for liquid rainfall conditions (frost-free periods), for which the reference measurements of the canopy water balance approach (WB) are reliable. Additionally, only events with footprints fitting inside the model domain were selected." (page 7: line 210)

**Page 10: line 283**

Typical situations occurring under rain or wet canopy conditions have been addressed in the data processing, such as potential signal loss of water vapor fluxes due to tube attenuation or sensor separation Fratini et al. (2012) and the detection of records with weak variance during stable conditions or low wind speeds (Vickers and Mahrt, 1997).

**Page 11: paragraph "conditions for implausible turbulent fluxes"**

Why should interception depend on rH. I can see that a correlate is with wind speed. Where is that considered? I wonder why all of the emphasis is on rH as a criterion here. Would'nt a canopy air temperature difference, or maybe a dew point depression be a variable more aligned with that fuxes?

We agree that a similar analysis as with rH in this chapter would be interesting with other flux aligned variables such as wind speed or temperature gradients. However, potential underestimation of LE during conditions of high relative humidity due to low pass filtering is a known issue with some (closed path) EC systems (Massman and Ibrom, 2008; Zhang et al., 2023). The analysis of meteorologic drivers during rainfall interception in figure 4 of the manuscript indicated that high relative humidity coincides with precipitation and wet canopy conditions. This was also shown in figure 5 b) of the manuscript additional to an increasing energy imbalance (decreasing LER) with increasing rH in figure 5 a). Hence, we decided to further investigate on the underestimation of LE by analysing the absolute and relative changes of  $LE_{EC}$  for different correction methods under these conditions.

The authors are pretty sure that their "physically based approach" is preferable to the machine learning approach, but their reliance on RH instead of more "flux related" variable sets suggests that they carry along a portion of the AI "black box" leanings. Please respond.

Our correction approach does not rely or is not related directly to rH as for example the approach of Zhang et al. (2023). Our approach substitutes  $LE_{EC}$  with the modelled  $LE_{2D}$  for events of interception and applies a Bowen ratio based correction for the remaining "dry" conditions. We used the rH dependent analysis of correction methods to allow comparability with the approach to Zhang et al. (2023).

**Some general questions for the authors to reply to:**

**1. How do the authors verify that current interception estimates are inadequate?**

We used the classical canopy water balance approach WB to calculate interception as the residual of measured liquid gross precipitation Pg and net precipitation Pn (the sum of free throughfall and drainage) collected by two gutters of 10m length with a total collection area of  $3.18m^2$ . A direct comparison of evaporation by different methods such as eddy covariance EC or canopy water balance WB is challenging due to different source areas and uncertainties in the respective approaches. The evaluation of the results of both methods assumes on the one hand similar or homogeneous interception properties in the respective source areas. On the other

hand, it is also assumed that transpiration and evaporation from litter/soil are negligible for saturated conditions or sufficiently closed canopies. Then, measured total evaporation  $E_{tot}$  by the EC approach can be substituted by evaporation of intercepted rain  $E_{\rm I}$  as done in our previous work in Fischer et al. (2023).

Hence, we used a 2D Rutter model approach recognizing the horizontal variability of the vegetation with a resolution of 10m in a gridded domain to account for the different source areas of the WB approach and the EC measurements. We demonstrated that the 2D Rutter model approach can reproduce sums of interception for the source area of independent canopy water budget measurements WB. Thus, we assume that the model reproduces sums of interception and total evaporation for the respective footprint areas of the EC measurements and can accordingly be used as a verification tool.

"Consequently, we combined the two methods to arrive at a consistent dataset adjusted for dry and wet conditions. This new dataset  $LE_{WB}$  incorporates the canopy water budget into the common practice to allocate the energy balance residual to the turbulent fluxes, in our case by preserving the Bowen ratio. First,  $LE_{EC}$  was replaced by modelled data  $LE_{2D}$  for interception conditions. The remaining "dry" dataset was then corrected on an half-hourly basis with the Bowen ratio based energy balance adjustment for daytime conditions after (Mauder, 2013)". (page 13: line 382-386)

**2. How are known issues with the eddy flux approach during rainfall addressed in this standardization?**

Known issues leading to faulty eddy flux measurements during precipitation are not addressed in our approach as they have not yet been clearly identified. Our approach simply detects intervals with interception, for which we have demonstrated a systematic underestimation of  $LE_{EC}$  as compared to independent measurements and quantified by the latent energy ratio LER. Then, these wet conditions are replaced by the model results.

We synchronized the time steps between modelled and measured data (2D approach and EC) for comparison. We identified issues with the eddy flux approach during precipitation and interception by filtering the EC data for modelled interception events (Pg > 0 and/ or modelled canopy storage C > 0). For these conditions, we substituted the eddy flux data with the model results. The Bowen ratio based correction (for daytime conditions) was then applied for the remaining "dry" dataset and resulting outliers were removed using the  $4\sigma$ -filtering method. The final dataset, corrected for dry and interception conditions, was then gapfilled by the use of the software package ReddyProc (Wutzler et al., 2018). Hence, the eddy flux data during rainfall and interception is adjusted by the 2D canopy model to account for a closed water and energy budget.

**3. How many actual observations (not "gap-filled") are in these Fluxnet evaporation data series during and after rainfall?**

The period 2008 to 2010 shows an above-average annual precipitation sum of  $1088\pm138$ mm as compared to the long-term record for the period 1991 to 2020 with an average sum of 842mm a-1. Interception conditions prevail on around  $55\pm7\%$  of all days of the year, of which  $21\pm3\%$  are with precipitation. Hence, a majority of data is affected by the systematic underestimation effect of LEEC during interception.

The data series for LEEC from 2008 to 2020 in half-hourly resolution consists of 52561 data points of which 2058 data points (3.9%) are missing. 68.4% of the gaps are located within interception events, of which 28% occur during rain conditions ( $P_g>0$ ) and 71.6% during conditions with water stored on the canopy (C>0).

Similar to our previous paper are only 24% of the LE measurements during rain conditions flagged as data of good quality, while 46.3% are of moderate quality (flag=1) and 28.3% are flagged as data of bad quality (flag=2). During dry conditions (no rain and a dry canopy), 54.1% of the LE measurements are of good quality (flag=0), 30.8% of moderate and only 15.2% of bad quality.

4. The authors state that intercepted rain is returned to the atmosphere 'immediately after' the rainfall, but this is not the case with nocturnal rain, for which intercepted water may not clear the canopy until the follows day's convective surface layer reestablishes. Czikowsky and Fitzjarrald (2009) found that rainfall intercepted during nocturnal rainfall evaporated during the first convective hours of the following morning. How about that? Please comment how this is dealt with. This has to do with how one decides what part of the subsequent LE 'belongs' the overnight rainfall.

Sorry, this is a misunderstanding, thank you for highlighting it.

Processes and model parameters such as partitioning of precipitation and evaporation components, drainage or canopy storage capacity, are calculated in the 2D Rutter approach as a function of PAI. Storage depletion is simulated by an exponential drainage approach and by evaporation, which is calculated based on the Penman-Monteith equation. Thus, ventilation and energy availability are driving evaporation and how fast the intercepted rain returns into the atmosphere. The following figure shows an example event in September 2008, with a total duration of 27 hours from the first pulse of precipitation until the canopy storage C is completely emptied (dry canopy). The duration of the interception event is long since the last precipitation pulse occurs between 16:30 and 17:30 CET where energy availability is low. During the night with zero net radiation, evaporation rates are low, and the canopy storage depletion is slow as the only drivers are wind u and a relatively small vpd. During the first daylight hours at the next day around 6:00 CET, net radiation and vpd are increasing which leads to increasing rates of total evaporation and a strong decrease of water stored on the canopy.

Figure 1: Meteorologic drivers, water and energy budget related components for the onset of an interception event at 12:30 CET at first of September 2008. The panels show (from top to bottom) precipitation Pg, modelled canopy water storage C, vapor pressure deficit, horizontal wind velocity u, net radiation Rn and turbulent fluxes LE and H in half hourly resolution. Turbulent fluxes are shown as measured by the EC system (solid line) and modelled by the 2D Rutter approach (dot-dashed line).

**5. Can the authors address the issue of interception as a function of rainfall rate?**

Besides the canopy structure, which is considered in the 2D model as the horizontally variable PAI (see point 4), precipitation regime is another important factor affecting canopy interception. Figure 2 shows for each selected (liquid) interception event how the fraction of rainfall intercepted by the canopy surface  $(E_I:P_g)$  decreases asymptotically with increasing precipitation totals.  $E_I:P_g$  is highest for small events up to precipitation totals of around 10 mm of which 58% are captured by the canopy and evaporate back to the atmosphere. With higher rainfall totals, more water leaves the canopy storage as drainage and a lower fraction remains as intercepted rainfall. For large events exceeding a precipitation amount of 35 mm, the fraction decreases to less than 20%.

Figure 2: Asymptotic relationship between the fraction of rainfall intercepted by the canopy and total rainfall for selected liquid interception events

Figure 3 shows a similar relationship for evaporation totals of intercepted rainfall and the event's maximum rainfall intensity  $PI_{max}$ . The figure only considers events less than 10h to exclude events with high totals of  $E_{\rm I}$ , which are the result of long but less intense precipitation. The graph shows that short events with a rainfall intensity above 8 mm h-1 lead to a quasi constant amount of  $E_{\rm I}$  of about 4 mm.

Figure 3: Relationship between evaporation of intercepted rainfall and maximum rainfall intensity for short interception events with a duration less than 10 hours

6. Do the authors only use the Vaisala RH sensor to determine this quantity? Is there evidence that it reports as accurately as promised by the manufacturer near saturation conditions? Many such instruments do not do well then, and sometimes manufacturers overlook this issue, especially when spec sheets are made. Over what RH range do you trust the sensor? Can it compared with observations from other instruments on site?

Thanks for addressing this. We are aware of the shifts in the capacitance sensors output (like VAISALA, HumiCap etc.) as is ICOS. The ICOS protocol includes scheduled calibrations and trend corrections. Regardless, raw outputs could sometimes exceed 100%. If sensors age too much, they are replaced. Additional humidity measurements include a gradient measurement with a gas analyser at the tower and psychrometer measurements as well as dew point sensors in the lab.

7. Do you think a similar approach could work at the Norway spruce conifer forest in Europe considered in this paper? Question is related to the paper (Czikowsky and Fitzjarrald, 2009):

LBA-ECO. Event-based ensembles were found to estimate the eddy fluxes LE and H above a forest in Brazil. The basic eddy covariance data was recalculated in such a way that measurements were filtered to continue returning values during rainfall, and, re-starting the calculation at the moment the rain stopped, flux calculation periods were begun. Consider two "treatments" over the same forest, first a base state ensemble: days without rain, under the same radiative conditions as rain days. The second group is a precipitation event ensemble. This approach works well in the regular weather patterns of the tropics; perhaps it would be problematic in Europe. It is incorrect to state the this approach has no error estimate, as has appeared in the literature.

The method developed by Czikowsky and Fitzjarrald (2009) is essentially based on correct eddy covariance measurements. As already shown in Fischer et al. (2023) and this study, eddy measurements underestimate evaporation during periods of wetting. This means that, according to the measurements at our site, the method developed by Czikowsky and Fitzjarrald (2009) would lead to an underestimation of interception.

A similar approach considering two treatments was applied in Lian et al. (2022) who used "physics-informed hybrid machine learning models built under wet versus dry conditions". Data of poor quality was excluded and only liquid rainfall conditions were considered. Additionally, they corrected the data for the systematic underestimation of  $LE_{EC}$  during interception by accounting for the dependence of latent energy ratio LER on relative humidity by using a neural network. This relationship was also highlighted in our paper.

So, if the "wet" treatment is corrected for this potential uncertainty, the approach of Czikowsky and Fitzjarrald (2009) could be applied at the Norway spruce conifer forest site, although it requires a long data set, since the "wet" treatment contains more poor quality data and the systematic underestimation during interception events needs to be accounted for. Regardless, problems may arise from any major shift in atmospheric conditions that is not adequately covered with data of acceptable quality. So, such an approach remains to be tested. We also agree to the remark regarding

the prevailing climate and the canopy structure. Both will affect the outcome and the need of evaporation corrections for interception.

**References**

[revised manuscript text omitted]

---

## Author Comment (AC2)

Thank you very much for the constructive review of our manuscript, also for appreciating this work and supporting to improve it by valuable remarks and suggestions within the manuscript. We carefully went through all the comments and revised the manuscript accordingly. Replies to comments relating to specific lines, paragraphs or tables have been incorporated accordingly. Since the upload of the revised manuscript is not possible during this stage of publishing, revised sections are given by page and line number. Additionally, we address specific questions and major points of the reviewer below.

**Reviewer #2:**

Terminology / notation pass: Define/contrast LAI vs PAI early (site LAI of 7.1 near the tower vs domain-mean PAI of 4.65 used by the model) and maintain consistent symbols; a reader-aid table would help.

Thank you for the valuable input. The following was added in the method section describing the "2D Rutter Model" and the spatial distribution of the PAI in the study domain (Figure 1):

"The resulting spatially variable PAILocal of the study domain is presented in figure 1 with an overall PAI of  $4.65 \text{m}^2 \text{m}^{-2}$  and a PAI of  $6.54 \text{m}^2 \text{m}^{-2}$  for the area of the gutter measurements used for the canopy water balance approach. Please note that this values are somewhat smaller than the LAI given in the description of the study site, as they are i) methodologically different measures and ii) not related to the same source area. The LAI of 7.1 around the measurement tower refers to the total projected leaf area per unit ground area and is a result of continuous in-canopy radiation measurements during the year 2008. The PAI includes both leaves and woody components and was derived from the above mentioned 3D representation of the forest." (page 7: line 184-190)

Winter caveat in the main text: You already flag that LE\_WB can exceed LEEB in Jan/Dec and that snow isnt explicitly handled. Consider a one-sentence caveat in the Abstract or Conclusions to prevent over-generalization?

Thank you, this is a very valuable suggestion. We added that the evaluated results refer to liquid rainfall conditions for which our approach "provides appropriate evaporation rates from intercepted liquid precipitation for the analyzed forest ecosystem" (page 1: line 13).

In the discussion we are also mentioning the limitation of our study:

"However, the model was only validated for liquid rainfall conditions and frost-free periods, since throughfall measurements are only reliable during these conditions. The application of the results to the whole year, especially situations with snowfall, should be further investigated. Firstly, there is a lack of reference data and secondly, the modelling approach does not differentiate between solid and liquid precipitation.

We expect that our combined water and energy balance adjustment approach  $LE_{WB}$  is still plausible, since snow interception for DE-Tha is estimated less than 2% if distinguishing these processes (Vorobevskii et al., 2022)." (page 16: line 501)

Share code? If feasible, provide a repository link for the 2-D Rutter implementation (Appendix A) and the footprint-weighting workflow to accelerate adoption.

The source code of the 2-D Rutter approach is available here <a href="https://github.com/Ron-Q/CanWat">https://github.com/Ron-Q/CanWat</a> and another description can be found here: <a href="https://doi.org/10.13140/RG.2.2.33956.59529">https://doi.org/10.13140/RG.2.2.33956.59529</a>.

the footprint calculation we used the R code from http://footprint.kljun.net/index.php in a slightly modified form. However, we would prefer not to publish the changes in the source code, as these obstacles may have been deliberately introduced by Natasha Kljun to prevent the code from being used without careful consideration. An application example for the calculation of the mask is added to Git repository the (https://github.com/Ron-Q/CanWat/blob/main/Sub/auxiliary/footprint 2D short.R).